# Similarities and Contrasts in Time-Mean Striated Surface Tracers in Pacific Eastern Boundary Upwelling Systems: The Role of Ocean Currents in Their Generation

**Ali Belmadani** [1,2,*] **, Pierre-Amaël Auger** [1,3,4] **, Nikolai Maximenko** [5] **, Katherine Gomez** [1,3,6] **and Sophie Cravatte** [7]

1   Department of Geophysics, University of Concepcion, Concepcion 4070386, Chile
2   Météo-France, Direction Interrégionale Antilles-Guyane, 97200 Fort-de-France, Martinique, France
3   Millennium Institute of Oceanography (IMO), University of Concepcion, Concepcion 4070386, Chile; katherine.gomez@imo-chile.cl
4   University of Brest, CNRS, IRD, Ifremer, Laboratoire d'Océanographie Physique et Spatiale (LOPS), IUEM, Brest 29280, France; pierre-amael.auger@ird.fr
5   International Pacific Research Center, School of Ocean and Earth Science and Technology, University of Hawaii at Manoa, Honolulu, HI 96822, USA; maximenk@hawaii.edu
6   School of Marine Sciences, Pontifical Catholic University of Valparaiso, Valparaiso 2340000, Chile
7   Laboratoire d'Etudes en Géophysique et Océanographie Spatiale, Université de Toulouse, CNES, CNRS, IRD, UPS, 31400 Toulouse, France; sophie.cravatte@legos.obs-mip.fr
*   Correspondence: ali.belmadani@meteo.fr

**Abstract:** Eastern boundary upwelling systems feature strong zonal gradients of physical and biological properties between cool, productive coastal oceans and warm, oligotrophic subtropical gyres. Zonal currents and jets (striations) are therefore likely to contribute to the transport of water properties between coastal and open oceanic regions. For the first time, multi-sensor satellite data are used to characterize the time-mean signatures of striations in sea surface temperature (SST), salinity (SSS), and chlorophyll-a (Chl-a) in subtropical eastern North/South Pacific (ENP/ESP) upwelling systems. In the ENP, tracers exhibit striated patterns extending up to ~2500 km offshore. Striated signals in SST and SSS are highly correlated with quasi-zonal jets, suggesting that these jets contribute to SST/SSS mesoscale patterns via zonal advection. Striated Chl-a anomalies are collocated with sea surface height (SSH) bands, a possible result of mesoscale eddy trains trapping nutrients and forming striated signals. In the ESP, the signature of striations is only found in SST and coincides with the SSH bands, consistently with quasi-zonal jets located outside major zonal tracer gradients. An interplay between large-scale SST/SSS advection by the quasi-zonal jets, mesoscale SST/SSS advection by the large-scale meridional flow, and eddy advection may explain the persistent ENP hydrographic signature of striations. These results underline the importance of quasi-zonal jets for surface tracer structuring at the mesoscale.

**Keywords:** striations; satellite data; sea surface temperature; sea surface salinity; chlorophyll-a; eastern boundaries; Pacific Ocean



## 1. Introduction

Eastern boundary upwelling systems (EBUS), such as those found along the Californian and Chilean coasts facing the eastern North and South Pacific subtropical gyres (Figure 1a), are unique regions known for their cool surface waters, extreme biological productivity, and key influence on climate [1,2]. In sharp contrast, the regions located further offshore are warmer and comparatively deprived of marine life.

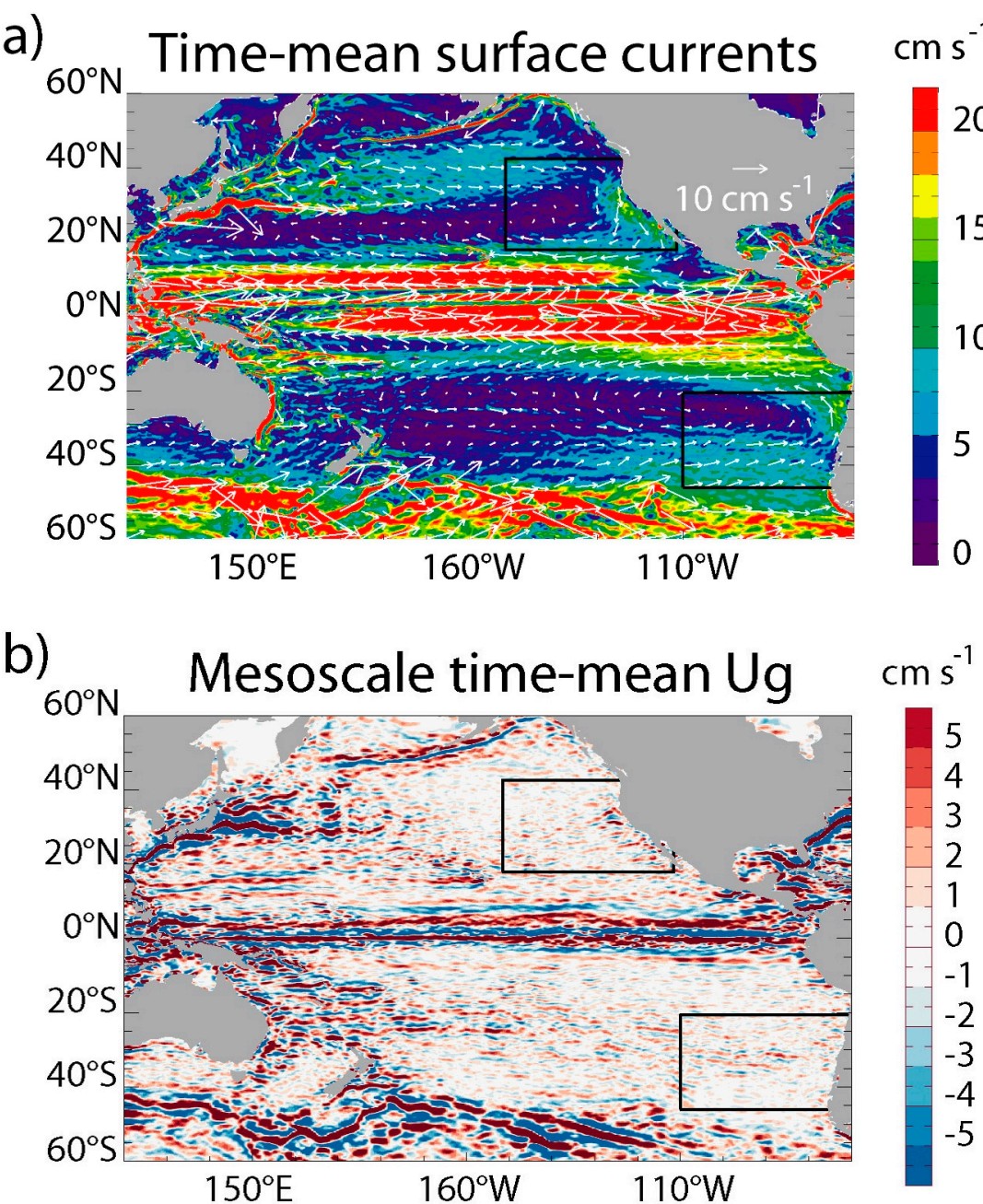

**Figure 1.** Surface circulation and quasi-zonal jets in the Pacific Ocean: (**a**) GlobCurrent surface current velocity (shading, cm s$^{-1}$) and vectors, and (**b**) SSALTO/DUACS spatially high-pass filtered zonal geostrophic velocity U$_g$, averaged over 2 July 2012–31 December 2018 (see Section 2 for a detailed description of the data and filtering method). The black boxes outline the two study regions off the Californian and Chilean EBUS in the subtropical North and South Pacific, respectively.

The oceanographic processes that exchange water between EBUS and neighboring oceanic regions are of particular interest because they may drive strong spatio-temporal variations in marine ecosystems and climate. Among such processes, mesoscale eddies are known to actively participate in the cross-shelf transport of ocean properties [3,4]. Other structures that potentially contribute to these exchanges but have received less attention are striations, which are multiple quasi-zonal jets in the long-term average circulation that have recently been discovered throughout the world's oceans and particularly in subtropical EBUS, using satellite, in situ, and model data [5–10] (Figure 1b). These bands of alternating eastward/westward mean geostrophic currents have meridional scales of 3–5°, extend from the surface to the intermediate or deep ocean, and spread out zonally for

1000 s of km [11]. Whether striations are quasi-stationary or transient features is a matter of debate: according to some authors, they are quasi-permanent structures [5,12–16], while others claim they are low-frequency modes of the oceanic mesoscale circulation exhibiting a slow drift, usually along the meridional axis [6,17–27]. Distinct phenomena occurring simultaneously and/or in different regions may be one explanation for such different interpretations. For instance, stationary or propagating striations may be extracted from the same data using either long or short averaging periods, while any stationary striations should be filtered out when considering velocity or sea level anomalies instead of the associated full fields [5,17].

Multiple theories have been proposed for their existence: artifacts of time-averaging westward-propagating mesoscale eddies [28–30]; beta-plumes generated through Rossby wave or eddy radiation from eastern boundary current meanders [15,16,31]; triad Rossby wave interaction and associated breaking [32–34]; anisotropic inverse energy cascades in geostrophic turbulence [35]; potential vorticity staircases associated with heterogeneous isopycnal mixing [36,37]; and topographic influences on potential vorticity [20,21,24], among others. Some of these theories were partly or entirely developed for EBUS [15,16,29,31].

In the Pacific subtropical EBUS, striations extend from the coast of California in the eastern North Pacific (ENP), but emerge farther offshore of Chile in the eastern South Pacific (ESP) (Figure 1b, see also Figures 2g and 3g), reflecting different generation mechanisms. California Current meanders anchored by coastline geometry may provide vorticity sources, which generate Rossby waves/mesoscale eddies that propagate westward [15,38]. Similar processes have been invoked in the ESP, except for vorticity sources driven by the topographic steering of gyre flows over deep-ocean escarpments rather than by eastern boundary current instabilities [16].

**Table 1.** Zonal and meridional wavelengths ($L_x$, $L_y$), and the striation angle ($\alpha$) associated with surface tracers and the dynamical fields determined from 2-D FFT in the ENP (Figure 4). For dynamical fields over the SST/Chl-a record (2012–2018), values on the left/right correspond to the black dashed boxes in Figure 2a/Figure 2e, respectively. Positive/negative $\alpha$ values are clockwise/counterclockwise.

| Variable | U | | $U_g$ | | SSH | | SST | SSS | Chl-a |
|---|---|---|---|---|---|---|---|---|---|
| Period | 2012–20218 | 2015–2018 | 2012–2018 | 2015–2018 | 2012–2018 | 2015–2018 | 2012–2018 | 2015–2018 | 2012–2018 |
| $L_x$ (°) | 14.2/9.8 | 18.3 | 18.3/9.8 | 18.3 | 18.3/7.5 | 18.3 | 18.3 | 14.2 | 6.1 |
| $L_y$ (°) | 2.8/3.1 | 3.0 | 2.8/3.1 | 3.1 | 3.0/3.9 | 3.3 | 3.0 | 3.7 | 3.3 |
| $\alpha$ (°) | −11.3/−17.6 | −9.2 | −8.8/−17.6 | −9.7 | −9.2/−27.3 | −10.2 | −9.2 | −14.4 | −28.3 |

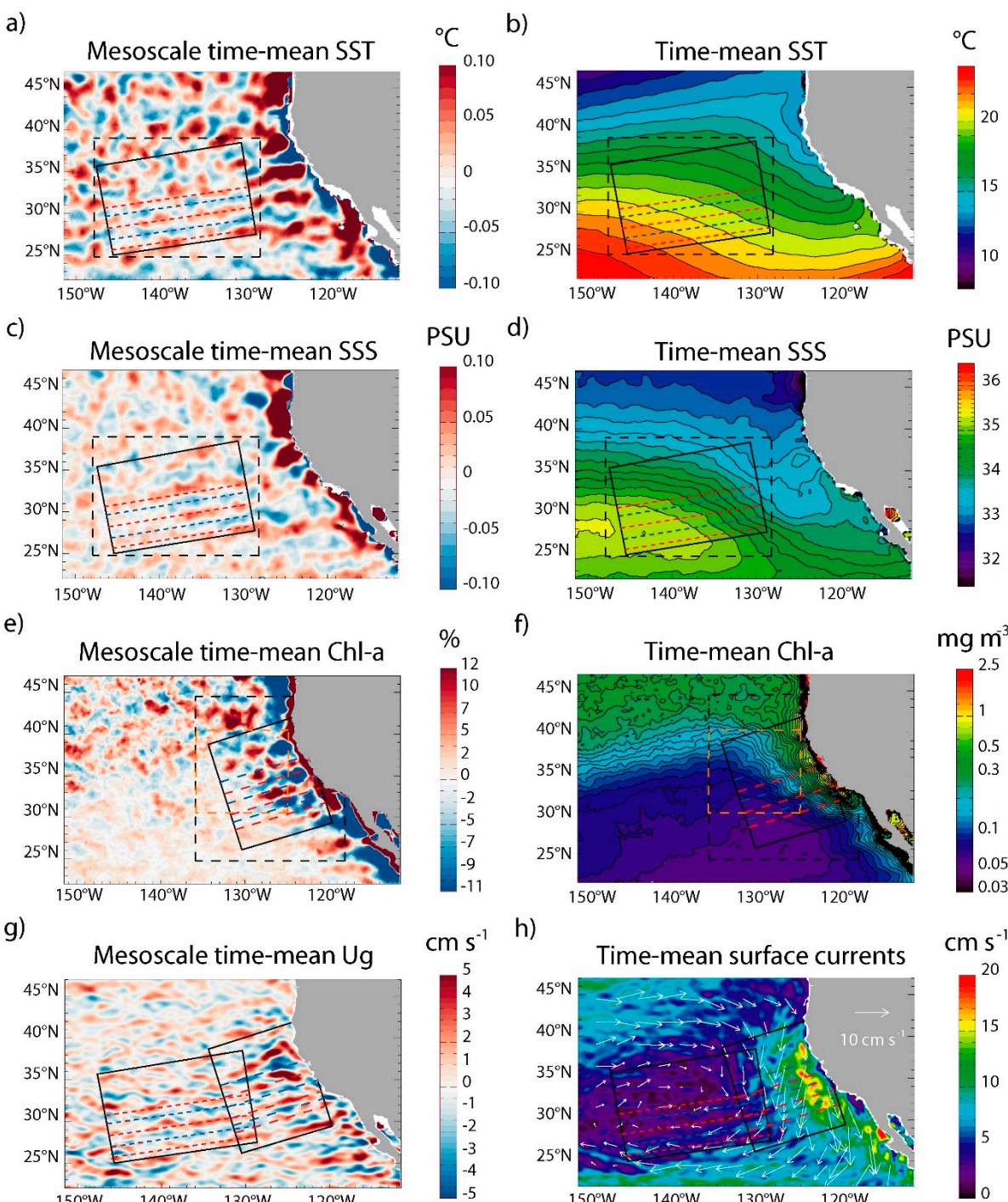

**Figure 2.** Striation expressions in physical/biological tracers and currents in the ENP: mean (**a**,**c**,**e**,**g**) spatially high-pass filtered and (**b**,**d**,**f**,**h**) full (**a**,**b**) AMSR-2 SST (°C), (**c**,**d**) SMAP SSS (PSU), and (**e**,**f**) GlobColour Chl-a; (**g**,**h**) mean (**g**) spatially high-pass filtered SSALTO/DUACS $U_g$ (cm s$^{-1}$) and (**h**) GlobCurrent full surface current velocity (shading, cm s$^{-1}$) and vectors. Chl-a units are mg m$^{-3}$ for (**f**), the full field, and % of the latter for (**e**), the high-pass filtered field (Appendix A). The black dashed boxes are where 2-D FFT is applied to the tracer fields and $U_g$, except for (**e**,**f**) Chl-a, for which the orange dashed box is used. The solid boxes are tilted with the striation angle in SSALTO/DUACS $U_g$ determined from 2-D FFT. The associated tilted red and blue dashed lines on all the panels indicate the approximate locations of eastward and westward jets, respectively, as inferred from (**g**) SSALTO/DUACS $U_g$. The 2 July 2012–31 December 2018 averaging period is considered throughout, except (**c**,**d**) where the 4 April 2015–31 December 2018 period is used due to the shorter SSS record, resulting in a slightly larger striation tilt (9.7° vs. 8.8°, see Table 1).

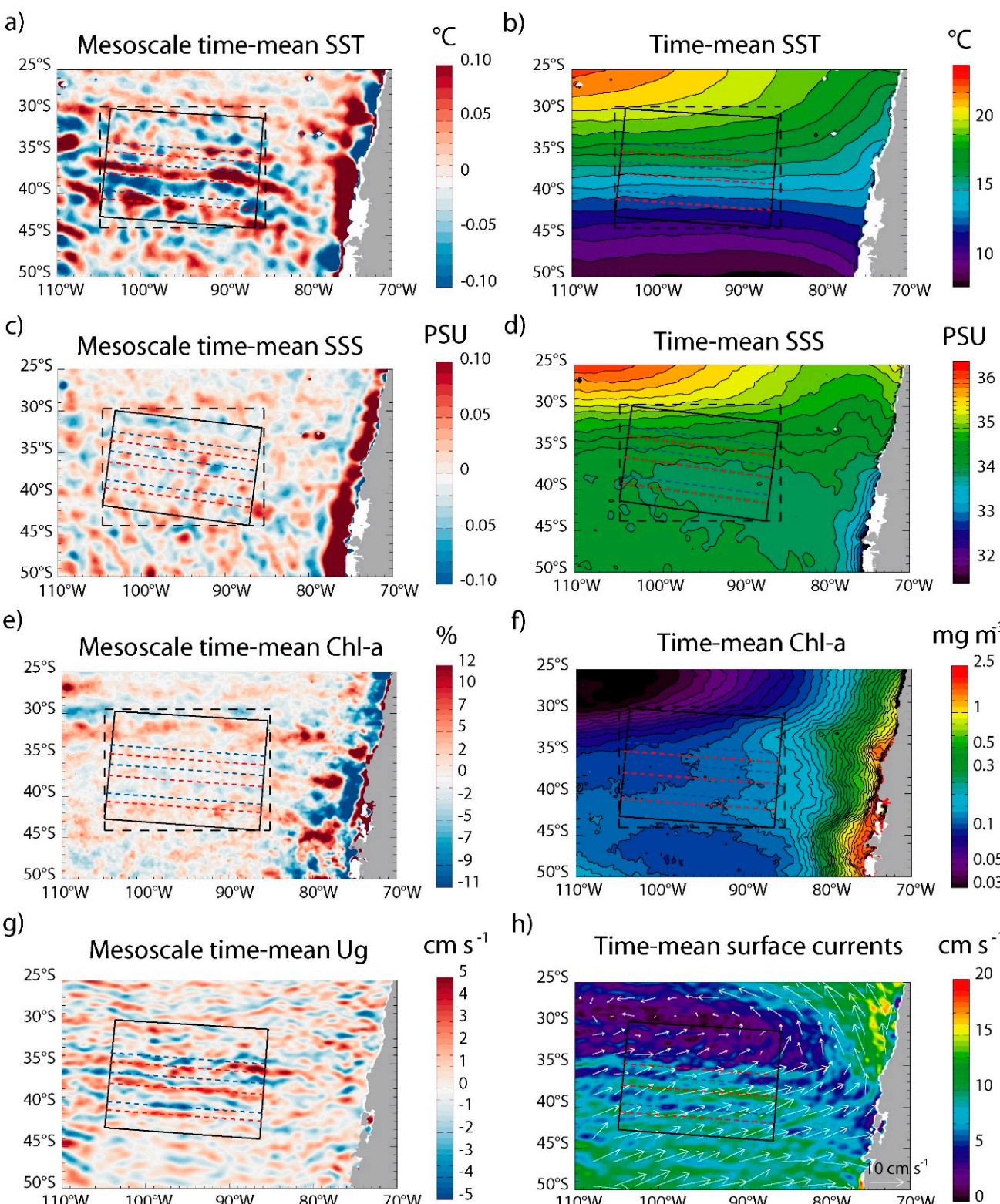

**Figure 3.** Same as Figure 2, except for the ESP. 2-D FFT is applied inside the black dashed boxes for all the variables. The striation tilt is larger over the 4 April 2015–31 December 2018 averaging period compared to the 2 July 2012–31 December 2018 period (12.7° vs. 5.8°, see Table 2).

**Table 2.** Zonal and meridional wavelengths ($L_x$, $L_y$), and the striation angle ($\alpha$) associated with surface tracers and the dynamical fields determined from 2-D FFT in the ESP (Figure 5). Positive/negative $\alpha$ values are clockwise/counterclockwise.

| Variable | U | | $U_g$ | | SSH | | SST | SSS | Chl-a |
|---|---|---|---|---|---|---|---|---|---|
| Period | 2012–2018 | 2015–2018 | 2012–2018 | 2015–2018 | 2012–2018 | 2015–2018 | 2012–2018 | 2015–2018 | 2012–2018 |
| $L_x$ (°) | 42.7 | 25.6 | 25.6 | 11.6 | 42.7 | 11.4 | 25.6 | 128 | 128 |
| $L_y$ (°) | 3.5 | 3.7 | 2.6 | 2.6 | 2.7 | 2.7 | 3.7 | 5.6 | 8.5 |
| $\alpha$ (°) | 4.6 | 8.1 | 5.8 | 12.7 | 3.7 | 13.2 | 8.1 | 2.5 | −3.8 |

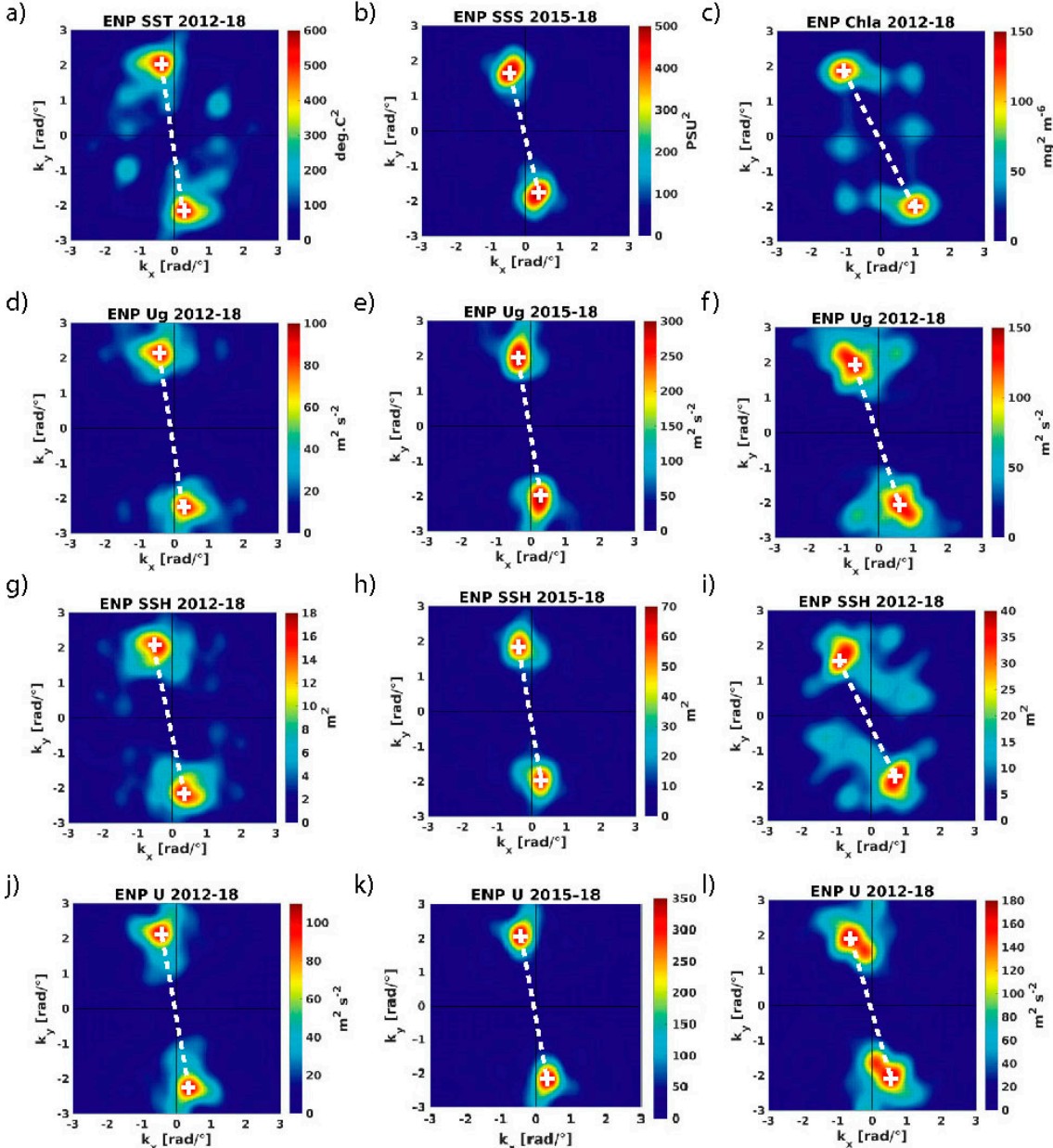

**Figure 4.** Spatial power spectra of mean spatially high-pass filtered fields in the ENP, computed within the dashed boxes in Figure 2. (**a**) AMSR-2 SST (°C$^2$); (**b**) SMAP SSS (PSU$^2$); (**c**) GlobColour Chl-a (mg$^2$ m$^{-6}$); (**d**–**f**) SSALTO/DUACS $U_g$ (m$^2$ s$^{-2}$); (**g**–**i**) SSALTO/DUACS SSH (m$^2$); and (**j**–**l**) SCUD U (m$^2$ s$^{-2}$). The data are averaged over (left and right columns) 2 July 2012–31 December 2018 and (middle column) 4 April 2015–31 December 2018 in (left and middle columns) the offshore region and (right column) the coastal transition zone. The white crosses indicate the locations of the symmetric maxima, joined by the dashed lines indicating the direction of the dominant wave vector.

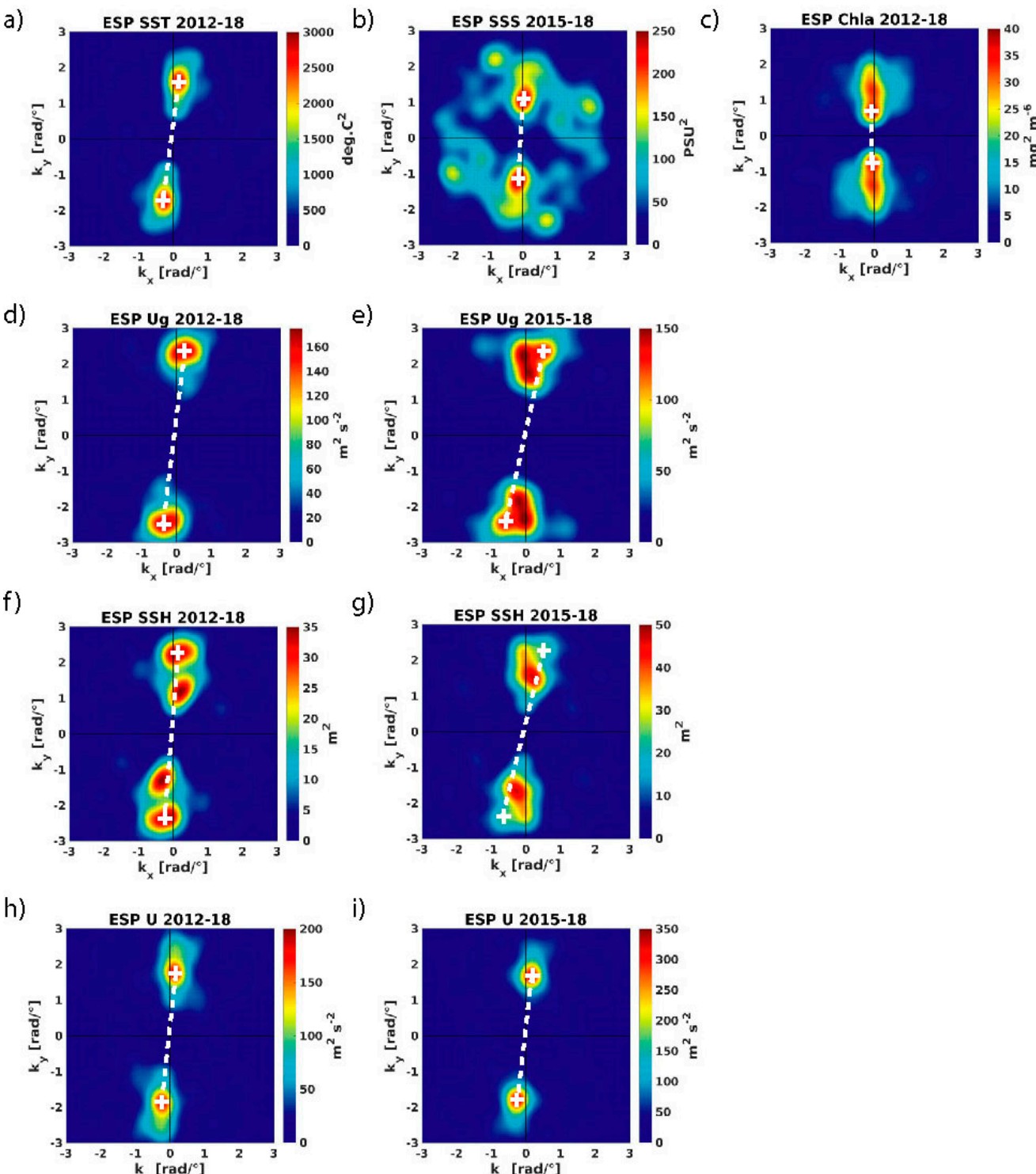

**Figure 5.** Spatial power spectra of mean spatially high-pass filtered fields in the ESP, computed within the dashed boxes in Figure 3. (**a**) AMSR-2 SST (°C$^2$); (**b**) SMAP SSS (PSU$^2$); (**c**) GlobColour Chl-a (mg$^2$ m$^{-6}$); (**d,e**) SSALTO/DUACS U$_g$ (m$^2$ s$^{-2}$); (**f,g**) SSALTO/DUACS SSH (m$^2$); and (**h,i**) SCUD U (m$^2$ s$^{-2}$). The data are averaged over (left column and c) 2 July 2012–31 December 2018 and (middle column) 4 April 2015–31 December 2018. The SST and Chl-a being analyzed over the same region and period, the corresponding U$_g$, SSH and U spectra are only shown once (**d–h**). The white crosses indicate the locations of the symmetric maxima (local maxima on (**e–g**)), joined by the dashed lines indicating the direction of the dominant wave vector (or wave vector consistent with striations).

Despite their weak amplitude (~1 cm/s), striations may advect the background temperature field [14], contribute to tracer mixing [37,39], ventilate oxygen minimum zones [40,41], transport floating debris [42], and modify surface winds [22], affecting marine ecosystems and climate. In particular, Ref. [39] showed from idealized modelling that striations may contribute much more to zonal mixing than eddies do when the background large-scale flow is predominantly zonal. The Lagrangian modelling study of Ref. [42] suggested the existence of an outflow from the ESP subtropical gyre dominated by striations, with its transport balancing 13–35% of convergence into the gyre. Moreover, Ref. [22]'s high-resolution coupled model study reported striated SST anomalies of $\pm 0.05 - 0.1\,^\circ$C off the coast of Peru in the tropical ESP as well as their impacts on the wind stress curl pattern. Considered together, these studies suggest that striations have a significant impact on water mass balances. Such effects may be further amplified in regions with strong along-striation (primarily zonal) physical and biogeochemical tracer gradients, such as the cool, highly productive subtropical EBUS.

However, a complete understanding of the physical processes leading to the impact of striations on water masses is missing. For instance, zonal advection of background sea surface temperature (SST) in regions of well-defined zonal gradients, like the tropical central/eastern South Pacific, has been invoked in Ref. [22]'s modelling study. Conversely, Ref. [5] reported subsurface temperature anomalies in phase with the sea surface height (SSH) signature of striations in the ENP and ESP EBUS from in situ and model data. Although they did not investigate the processes responsible for such an alignment, our interpretation is that it is inconsistent with zonal advection processes, such as the ones evidenced by Ref. [22]. Since the SSH signal of striations is in phase quadrature with geostrophic zonal jets [5], the striated mesoscale temperature field is also in phase quadrature with the geostrophic current. This, then, discards advection as the primary driver, which would otherwise require spatial correspondence between the mesoscale temperature and zonal velocity fields. On the other hand, Ref. [14] found that zonally-elongated satellite SST fronts were frequently associated with time-mean and transient striations i.e., quasi-zonal jets. However, they focused on the specific case of transient fronts within background meridional gradients and did not detail the relationship between long-term mean fronts and the associated striations, or the underlying dynamics. In this respect, Ref. [37] provided evidence of another mechanism responsible for striation impacts on water masses. They found staircase-like meridional profiles in tropical Pacific subsurface salinity and oxygen cruise data, with fronts/homogeneous regions inside eastward/westward jets, respectively. Such a structure was indeed hypothesized by Ref. [14] in the presence of meridional large-scale gradients, which characterize subsurface tropical Pacific salinity and oxygen [37]. This was also anticipated in the theories of Refs. [36,43]; however, whether these findings are relevant to the eddy-rich extra-tropics and different background properties (e.g., zonal gradients) remains unclear.

Multi-year satellite records are now available not only for such variables as SST or ocean color as a proxy for surface chlorophyll-a (Chl-a), but recently also for the less studied sea surface salinity (SSS), allowing us to grasp a broader picture of striation impacts on surface water masses. Here, we present observational evidence from multi-sensor data of striation effects on the aforementioned surface tracers in the North and South Pacific subtropical EBUS. The focus of this paper is on time-averaged, possibly quasi-stationary striations, as a first step to understanding their impacts on surface water properties. In addition to SST, for the first time, SSS and Chl-a reveal contrasted footprints of striations in these climatically and biologically important regions. The differences found among the patterns in different variables and regions are attributed to the differences among large-scale tracer distributions and the embedded striations. A detailed analysis of advection terms from surface tracer equations is used to assess the dynamics responsible for the existence and persistence of striated SST and SSS signals in the ENP. The influence of oceanic jet temporal variability is briefly discussed.

## 2. Materials and Methods

### 2.1. Satellite Data

Unless stated otherwise, the data are considered over a common period from 2 July 2012 to 31 December 2018 for the ENP (defined as 111.5° W–151.5° W, 22° N–47° N) and ESP (70° W–110° W, 25° S–50° S). The time span of the shorter SSS dataset is specified in Section 2.1.3.

#### 2.1.1. SSH and Currents

Altimetry data are used to characterize the imprint of striations in SSH and geostrophic currents (e.g., Figure 1b). We used the daily gridded (0.25°) SSALTO/DUACS DT2018 delayed-time level-4 merged 2-satellite product distributed by the Copernicus Climate Change Service. Absolute dynamic topography (i.e., SSH) and zonal geostrophic velocity ($U_g$), available from 1 January 1993 onwards (3 June 2020 at the time of writing), were extracted for the study period.

Total surface currents, including the Ekman flow are, however, required for the estimation of surface tracer advection (see Section 2.2). In particular, Ekman currents are expected to contribute significantly to the large-scale surface flow and associated advection. We used the daily gridded 0.25° GlobCurrent MULTIOBS_GLO_PHY_REP_015_004 level-4 product, available from 1 January 1993 to 31 May 2020 and distributed by the Copernicus Marine Service [44]. It combines the SSALTO/DUACS DT2018 altimetric geostrophic current and modeled Ekman surface current using wind stress from the European Centre for Medium-Range Weather Forecasts (ECMWF) ERA5 reanalysis [45].

Since the GlobCurrent and SSALTO/DUACS data are fully consistent, in terms of the geostrophic currents that should dominate mesoscale velocities, another product is needed to test the sensitivity of striation properties to the choice of dataset (see Section 2.2). For this purpose, near-surface currents derived from the altimeter SSH and scatterometer surface winds, and consistent with Lagrangian drifter trajectories, were obtained from the Surface CUrrents from Diagnostic model (SCUD) data available from APDRC/IPRC [46]. Daily gridded (0.25°) zonal (U) and meridional (V) velocities are publicly available from 14 February 2012 to 31 December 2018.

#### 2.1.2. SST

Following Ref. [14], remotely-sensed microwave data were used to extract the striated patterns in SST. We used 3-day average gridded (0.25°) Advanced Microwave Scanning Radiometer 2 (AMSR-2) [47] version 8, available from 2 July 2012 onwards. The UK Met Office Group for High Resolution Sea Surface Temperature (GHRSST) Operational Sea Surface Temperature and Sea Ice Analysis (OSTIA) [48] was also used for comparison over the AMSR-2 period. OSTIA uses infrared retrievals, which permits a higher 0.05° resolution but has a greater sensitivity to cloud cover. Importantly, it is completely independent from AMSR-2 and, therefore, suitable to assess the robustness of the striated SST signal.

#### 2.1.3. SSS

The SSS signature of striations was extracted from the NASA Soil Moisture Active Passive (SMAP) observatory [49], available from 4 April 2015 onwards. We used the JPL SMAP level-3 CAP standard mapped image 8-day running mean V4.3 validated 0.25° daily data from 4 April 2015 to 31 December 2018 [50]. SMAP was compared to the Soil Moisture Ocean Salinity (SMOS) data [51] over this time period. Although sharing the same horizontal resolution, the two datasets differ in terms of sensors, retrieval algorithms, and error corrections, among others [52].

### 2.1.4. Chl-a

The signature of striations in surface Chl-a was evaluated using satellite ocean color data. We used a merged product of the MODIS-AQUA and VIIRS datasets provided by the GlobColour project from July 2012 to December 2018. Monthly level-3 binned images (~25 km resolution) include all level-2 products accumulated over 1-month periods and merged using a bio-optical model-based procedure validated at the global scale [53].

### 2.2. Data Processing and Analysis

SSH and $U_g$ daily maps were time-averaged over the study period (2012–2018, see Section 2.1) and spatially high-pass filtered with a 4° half-width 2-D Hanning window to extract the signature of mean striations in the dynamical fields ([16], and references therein). Its sensitivity to the choice of the averaging period is discussed in Section 4.

The same procedure was applied to daily U, SST, SSS, and monthly Chl-a. In the latter case, logarithmic transformation was applied beforehand to reduce the strong gradient between the oligotrophic open ocean and the productive EBUS (see Appendix A). Such a transformation is commonly used for the analysis of Chl-a data (e.g., [54], their Appendix A, and references therein). Note that although permutating log-transformation and 2-D filtering does not theoretically lead to identical results, they are similar despite somewhat noisier signals than the ones presented in Section 3. In such a case, the computed large-scale component of Chl-a (which was removed to extract the mesoscale component, as for other variables [16]) was influenced by the strong near-zonal gradient, possibly leading to a less accurate extraction of other large-scale features, and therefore aliasing the extracted striated pattern. Applying the log-transformation first also allowed for the assessment of the impact of striations relative to background Chl-a (Appendix A). All the variables were filtered twice to remove any residual large-scale signal, which was significant for the SSH and tracers after the first filter application (not shown).

Following Ref. [16], 2-D Fast Fourier Transform (FFT) was applied to the previously obtained mesoscale mean fields, in order to compare the spatial scales of any striated patterns in the tracers and dynamical fields. For each variable and subdomain, FFT was computed in the area in which the striation expressions in the tracer and dynamical fields were most evident via a visual inspection (black dashed boxes in Figures 2 and 3). Such a method was, however, unsuccessful for ENP Chl-a because of the very strong near-meridional alongshore signal associated with the exceptional coastal productivity that prevented the extraction of the weaker near-zonal signals in the coastal transition zone (Figure 2e,f). Approximate spatial scales for the striated signal in Chl-a were then estimated from the FFT applied to a region that excluded the coastal band, while capturing part of the striated pattern (orange dashed box in Figure 2e,f). Note that the relevance of the visual identification of the areas used to apply FFT for the different variables is confirmed with an objective approach based on the velocity and tracer gradient anisotropy ([30,55], see Appendix B).

Besides zonal and meridional wavelengths, the average tilt of the striations from the zonal direction was derived from the energy spectra. The angle of the dominant wave vector connecting the symmetric, most energetic peaks was used for such a purpose (white dashed lines in Figures 4 and 5). In a few specific cases in the ESP ($U_g$ averaged over the shorter SSS record, and SSH for both time spans, Figure 5e–g), secondary maxima with meridional wavelengths consistent with those found for the single peak in $U_g$ (averaged over the SST record, Figure 5d) were used, instead. The angle obtained from the $U_g$ spectrum was used to average the mesoscale fields in the along-striation direction (solid boxes on Figures 2 and 3), allowing for the comparison of cross-striation profiles.

For each tracer, cross-correlation with the SSH and $U_g$ profiles was used to assess the possible role of zonal advection in striated pattern formation. Indeed, if the tracer is in phase with $U_g$ (and, therefore, in phase quadrature with SSH), then zonal advection should be involved. Conversely, if the tracer is in phase with SSH (and in quadrature with $U_g$), then zonal advection should not be involved, and some other mechanism can be suspected

(e.g., potential vorticity staircase [36,37,43]). If the phase lag is somewhere in between, then more than one mechanism, including advection, may be at play. Larger lags may either be indicative of inverse relationships, in which case the previously described cases apply after sign reversal, or poor correlation values. Interpolating profiles at a 0.01° resolution allows the computation of accurate lags. For each variable pair, the statistical significance at the 5% level of the correlation coefficient was computed using a Monte Carlo technique with 1000 iterations, whereby the first profile was correlated with a synthetic profile obtained using the Fourier analysis of the second profile and a random phase [56].

Finally, a decomposition of SST and SSS advection terms was computed in the ENP, where advection was found to play a dominant role (Section 3). However, as striations are associated with eddy trains in subtropical gyres [14,57] and particularly the ESP [16], both advection by the mean jets and nonlinear advection by individual eddies are candidate mechanisms. Moreover, dissipative processes should balance such advection in order to explain the persistence of the hydrographic signature of striations in multi-year averages (Section 3). The decomposition allows these questions to be addressed by separating near-zonal, along-striation advection from near-meridional, cross-striation advection; time-mean fields from anomalies relative to the mean (Reynolds decomposition); and large-scale components from mesoscale components for currents and hydrographical data.

The general equation of a surface hydrographical tracer $F$ (either SST or SSS) is:

$$\partial F / \partial t = - V \nabla F + \nabla (K \nabla F) + \text{external forcing} \tag{1}$$

where $t$ represents time, $V$ is the surface velocity vector, and $K$ represents diffusivity. The term on the left hand side represents the local (Eulerian) time derivative of $F$ (or tendency). It is driven by the three terms on the right hand side: from left to right, advection by ocean currents (both horizontal and vertical), turbulent mixing (horizontal and vertical), and fluxes at the ocean surface or along continental boundaries (such as heat/freshwater air–sea fluxes and river runoff).

In this study, we focused on the long-term mean striated patterns at the ocean surface, where horizontal advection is expected to dominate. In a near-Cartesian orthogonal frame, slightly rotated counterclockwise and aligned with mean striation axis, the horizontal advection of $F$ can be written as:

$$- V \nabla_H F = - u_a \partial F / \partial x_a - v_c \partial F / \partial y_c \tag{2}$$

where $\nabla_H$ represents the horizontal advection of $F$, $u_a$ and $v_c$ are the along-striation and cross-striation components of $V$, and $x_a$ and $y_c$ are the along-striation and cross-striation coordinates, respectively. The terms on the right hand side of (2) represent the along-striation and cross-striation advection of $F$.

A Reynolds decomposition was then applied to the surface currents and tracer:

$$u_a = u_a\prime + \overline{u_a} \tag{3}$$

$$v_c = v_c\prime + \overline{v_c} \tag{4}$$

$$F = F\prime + \overline{F} \tag{5}$$

where the overbars are for time-averaging over the available record, and the prime marks are for the anomalies relative to the time-averaged fields.

Averaging (2) over time yields the following:

$$- \overline{V \nabla_H F} = - \overline{u_a\prime \partial F\prime / \partial x_a} - \overline{u_a} \partial \overline{F} / \partial x_a - \overline{v_c\prime \partial F\prime / \partial y_c} - \overline{v_c} \partial \overline{F} / \partial y_c \tag{6}$$

The terms on the right hand side of (6) represent the long-term averages of nonlinear (or eddy) along-striation advection, mean along-striation advection, nonlinear cross-striation advection, and mean cross-striation advection, respectively.

Finally, spatial scale separation was applied to the anomalous and mean surface currents and tracer:

$$u_a\prime + \overline{u_a} = \mathrm{u}_{aL}\prime + \mathrm{u}_{aH}\prime + \overline{u_{aL}} + \overline{u_{aH}} \tag{7}$$

$$v_c\prime + \overline{v_c} = \mathrm{v}_{cL}\prime + \mathrm{v}_{cH}\prime + \overline{v_{cL}} + \overline{v_{cH}} \tag{8}$$

$$F\prime + \overline{F} = F_L\prime + F_H\prime + \overline{F_L} + \overline{F_H} \tag{9}$$

where *H* and *L* subscripts refer to the mesoscale and large-scale components obtained from spatial high-pass filtering with a 4° half-width 2-D Hanning window.

Incorporating (7), (8), and (9) into (6) gives the following:

$$
\begin{aligned}
-\overline{V\nabla_H F} = &-\left( \overline{u_{aL}\prime \partial F_L\prime/\partial x_a} + \overline{u_{aL}\prime \partial F_H\prime/\partial x_a} + \overline{u_{aH}\prime \partial F_L\prime/\partial x_a} + \overline{u_{aH}\prime \partial F_H\prime/\partial x_a} \right) \\
&-\left( \overline{u_{aL}} \partial \overline{F_L}/\partial x_a + \overline{u_{aL}} \partial \overline{F_H}/\partial x_a + \overline{u_{aH}} \partial \overline{F_L}/\partial x_a + \overline{u_{aH}} \partial \overline{F_H}/\partial x_a \right) \\
-&\left( \overline{v_{cL}\prime \partial F_L\prime/\partial y_c} + \overline{v_{cL}\prime \partial F_H\prime/\partial y_c} + \overline{v_{cH}\prime \partial F_L\prime/\partial y_c} + \overline{v_{cH}\prime \partial F_H\prime/\partial y_c} \right) \\
&-\left( \overline{v_{cL}} \partial \overline{F_L}/\partial y_c + \overline{v_{cL}} \partial \overline{F_H}/\partial y_c + \overline{v_{cH}} \partial \overline{F_L}/\partial y_c + \overline{v_{cH}} \partial \overline{F_H}/\partial y_c \right)
\end{aligned}
\tag{10}
$$

The 16 terms on the right hand side of (10) are the considered SST/SSS advection terms.

After 2-D mapping was performed, cross-striation profiles were computed and the key terms were cross-correlated with $\overline{F_H}$ as described above. An in-phase relationship between the tracer profile and some advection terms is indeed expected, considering that the right hand side of (1) can also be framed in terms of a balance between generation and dissipation processes. Using a simple Rayleigh dissipation model, (1) can be rewritten for the mesoscale tracer cross-striation profile as the following:

$$\partial < F_H > /\partial t = G - c < F_H > \tag{11}$$

where $<.>$ is for averaging along $x_a$ within the study region, *G* represents generation processes, $c = 1/t_0$ is a constant dissipation coefficient, and $t_0$ is the associated dissipation time scale. After averaging over long periods of time, $< F_H >$ tendency is assumed to vanish for time-mean striations, so that:

$$< \overline{F_H} > = t_0 \overline{G} \tag{12}$$

The striated tracer signal is thus expected to correlate with the processes responsible for its generation. The aforementioned correlation analysis can then be used to verify whether the advection terms of (10) can be considered as such generation processes. Any such term (or combination of terms) can also be used to estimate $t_0$, which indicates how fast the striated tracer pattern would be generated from a smooth background tracer field or, equivalently, how fast it would dissipate if the generation stopped. Such an estimate can be provided by the ratio of the standard deviations (along $y_c$) of the mesoscale tracer cross-striation profile $\langle \overline{F_H} \rangle$ and related advective generation term.

### 3. Results

#### 3.1. Striated Surface Tracers

Figure 2 shows the time-averaged, spatially filtered, and full (raw, unfiltered) satellite surface tracer and current observations in the ENP. Striated signals in SST and SSS are visible over a large region extending between the California coast and ~145° W (Figure 2a,c, see also Figure A1a,b). Despite the weak signals, with only a few local values of ± 0.05–0.1 °C and ± 0.05–0.1 PSU being statistically significant at the 5% level (see Appendix C), the banded patterns are zonally coherent and appear connected to stronger signals closer to the coast, suggesting they are likely real features. The statistical analysis of the profiles obtained after averaging the mesoscale tracer fields along the striation axis also supports this interpretation, as will be shown later. Moreover, OSTIA exhibited mesoscale SST structures almost identical to AMSR-2, except in the coastal region (Figure 2a and

Figure S1a, see also Figure A1a and Figure S2a), which further supports their robustness. On the other hand, mesoscale SSS structures from SMOS data are much noisier than SMAP, although some banded pattern can still be distinguished (Figure 2c and Figure S1b, see also Figure A1b and Figure S2b). Similar meridionally-alternating bands were found in Chl-a data with magnitudes reaching ~10% of full Chl-a, but they did not extend beyond ~500 km offshore (Figure 2e, see also Figure A1c).

Meridional wavelengths $L_y$ inferred from spectral analysis (Figure 4) ranged from 3.0° to 3.7° for different tracers, and striation axes were tilted counterclockwise from the zonal orientation by an angle $\alpha$ varying from 9.2° to 14.4° (28.9° for Chl-a, Table 1). These figures agree well with the dynamical fields (U, $U_g$, and SSH) having $L_y$ range from 2.8° to 3.9°, and $\alpha$ from 8.8° to 11.3° (27.3° for SSH in the region with striated Chl-a signals). The spectral properties of the banded tracer signals are therefore similar to those of the previously reported time-mean striations [5] shown on Figure 2g, suggesting a dynamical relationship between the former and the latter. Note that although any spectral peak aliasing by other hypothetical zonally-elongated mesoscale current/tracer features cannot be fully discarded, it is not expected to be critical: maximum power for small zonal/mesoscale meridional wavenumbers is consistent with both the visually inferred striations (Figure 2) and marked mesoscale current/tracer zonal anisotropy (Figure A1).

Remarkably, warm and salty (cool and fresh) anomalies were aligned with eastward (westward) jets (Figure 2a,c,g), further suggesting they may result from advection by the striated currents (see Section 2.2). Indeed, the region through which the jets extended south-westward has strong background along-striation gradients of SST and SSS (Figure 2b,d) between the Californian EBUS and the subtropical gyre (Figure 2h). In contrast, the striated geostrophic velocities tended to coincide with zero-crossings of Chl-a anomalies, i.e., bands of locally more (less) productive waters were collocated with a negative (positive) meridional shear of near-zonal jets, that is, along the troughs (crests) of the SSH signal associated with striations (Figure 2e,g). This latter result suggests that advection may not play a dominant role in the formation of the banded Chl-a pattern and, instead, that some other mechanism may be at play. Interestingly, however, the offshore extent of the striated Chl-a pattern corresponds to that of the background Chl-a gradient between coastal and open-ocean waters (Figure 2f). A possible role of water mass trapping by mesoscale eddies is discussed in Section 4.

The same analysis yields strikingly different results for the ESP. While multiple near-zonal SST bands of amplitudes reaching $\pm 0.1 - 0.2$ °C locally (and therefore statistically significant at the 5% level, Appendix C) are evident in a region far offshore central Chile (Figure 3a), the SSS and Chl-a patterns are noisier (Figure 3c,e). Alternate SST/SSS datasets provided similar results. Again, the mesoscale SST pattern from AMSR-2 is reproduced almost perfectly by OSTIA (Figure 3a and Figure S1c, see also Figure A2a and Figure S2c), while the mesoscale SSS pattern is just as noisy in SMOS data compared to SMAP (Figure 3c and Figure S1d, see also Figure A2b and Figure S2d). Only AMSR-2 and SMAP are considered in the remainder of the paper for SST and SSS, respectively.

Spectral analysis confirms that the mesoscale properties of the SSS and Chl-a fields are inconsistent with those of the dynamical fields (Figure 5). $L_y$ are larger in the former and $\alpha$ are smaller (Table 2), suggesting that striations may not have a clear impact on these tracers in this region. This is also consistent with the lack of clear zonal anisotropy in SSS and Chl-a signals from the ESP (Appendix B). Let us, however, note that, as is the case in the ENP, some kind of banded structure is detected for mesoscale Chl-a in the coastal transition region, although with reduced zonal extent and away from the most obvious quasi-zonal jets located farther offshore along the subtropical gyre poleward edge (Figure 3e,g,h, see also Figure A2c,d). On the other hand, the values found for SST are within the range of those extracted from the dynamical fields (Table 2). Furthermore, the striated signals in SST tend to be in phase quadrature rather than in phase with those in $U_g$ (Figure 3a,g). These findings support the interpretation that the striated SST may be dynamically related to striations, although probably not through advection.

Part of the differences observed from the ENP may reside in different jet locations relative to the large-scale tracer fields. As subtropical EBUS, both systems exhibit similar gradients of SST, SSS, and Chl-a (Figure 2b,d,f and Figure 3b,d,f). However, unlike the ENP (Figure 2g), ESP striations are observed farther offshore and do not extend eastward to the coast (Figures 3g and A2d), consistently with distinct formation processes [15,16]. As a result, ESP striations are found within weak zonal gradients of SST, SSS, and Chl-a (Figure 3b,d,f,g), preventing zonal advection processes from playing any leading role. On the other hand, this region features a strong meridional SST gradient between the subtropical gyre and higher latitudes. The phase quadrature between the striated signals in SST and $U_g$ then implies that eastward (westward) jets correspond to mesoscale meridional SST gradients that sharpen (flatten) the large-scale gradient, in qualitative agreement with Ref. [37] and the potential vorticity staircase dynamics [36,43]. Mesoscale gradients are, however, too weak ($<10^{-3}$ °C km$^{-1}$, ~1/10 of the large-scale gradient) to generate any significant staircase profile in raw SST.

To further support our findings and more objectively assess the relationship between surface tracers and either $U_g$ or SSH, variables were averaged in the along-striation direction and in the area where striated signals were most evident (solid boxes on Figures 2 and 3), before cross-striation profiles were computed (Figure 6). Most peaks of the SST (SSS) profile are statistically significant at the 5% level (see Appendix C) in the 27–32° N (30–34° N) latitude band for the ENP, and in the 35–43° S band for the ESP (for SST only). For comparison, there are only slightly more significant peaks for SSH and $U_g$ in both regions. This confirms that the banded mesoscale structure of hydrographic tracer fields is robust, even though most smaller-scale details of their 2D structure may not be, as discussed previously (Figures 2 and 3). Almost all peaks of the Chl-a profile are significant in the ENP, with much fewer robust peaks in the ESP (Appendix C), which is, again, consistent with our previous findings.

Lag-correlation analysis confirmed that near-periodic ENP SST and SSS were both highly (~0.8) and significantly correlated with $U_g$, with only a small lag of approximately 0.1–0.2° (Figure 6a,b). Correlations were also high (~0.9) and statistically significant with SSH, but with a larger lag (~0.6–0.7° latitude) in the opposite direction, consistently with both near-quadrature, as one might expect from the quadrature between $U_g$ and SSH, and with dominant periodicity $L_y$~3°. Conversely, Chl-a and SSH ($U_g$) exhibited a high, statistically significant negative correlation ~0.7–0.8 with meridional lag ~−0.25° (~+0.5°) (Figure 6c). In addition, correlation at zero lag was significant with SSH only and higher (−0.64) compared to $U_g$ (−0.40). This confirms that zonal advection is not the dominant driver for the signature of time-mean striations in Chl-a, unlike physical tracers.

In the ESP, SST and SSH were approximately in phase, particularly between 37° S and 41° S (Figure 6d) with a high (~0.8), statistically significant correlation. The previously inferred quadrature with $U_g$, particularly evident also from 37° S to 41° S, was confirmed with maximum correlation ~0.7 at lag −0.8°. On the other hand and consistent with the previous analyses, SSS and Chl-a appeared to be poorly correlated with SSH and $U_g$ (Figure 6e,f). In fact, SSS and Chl-a profiles did not exhibit clear periodicity, unlike SST and the dynamical fields. Note that these results did not change much when the striation angle from SSH instead of $U_g$ was used to compute the along-striation averages (not shown), consistently with the generally similar $\alpha$ values (Tables 1 and 2).

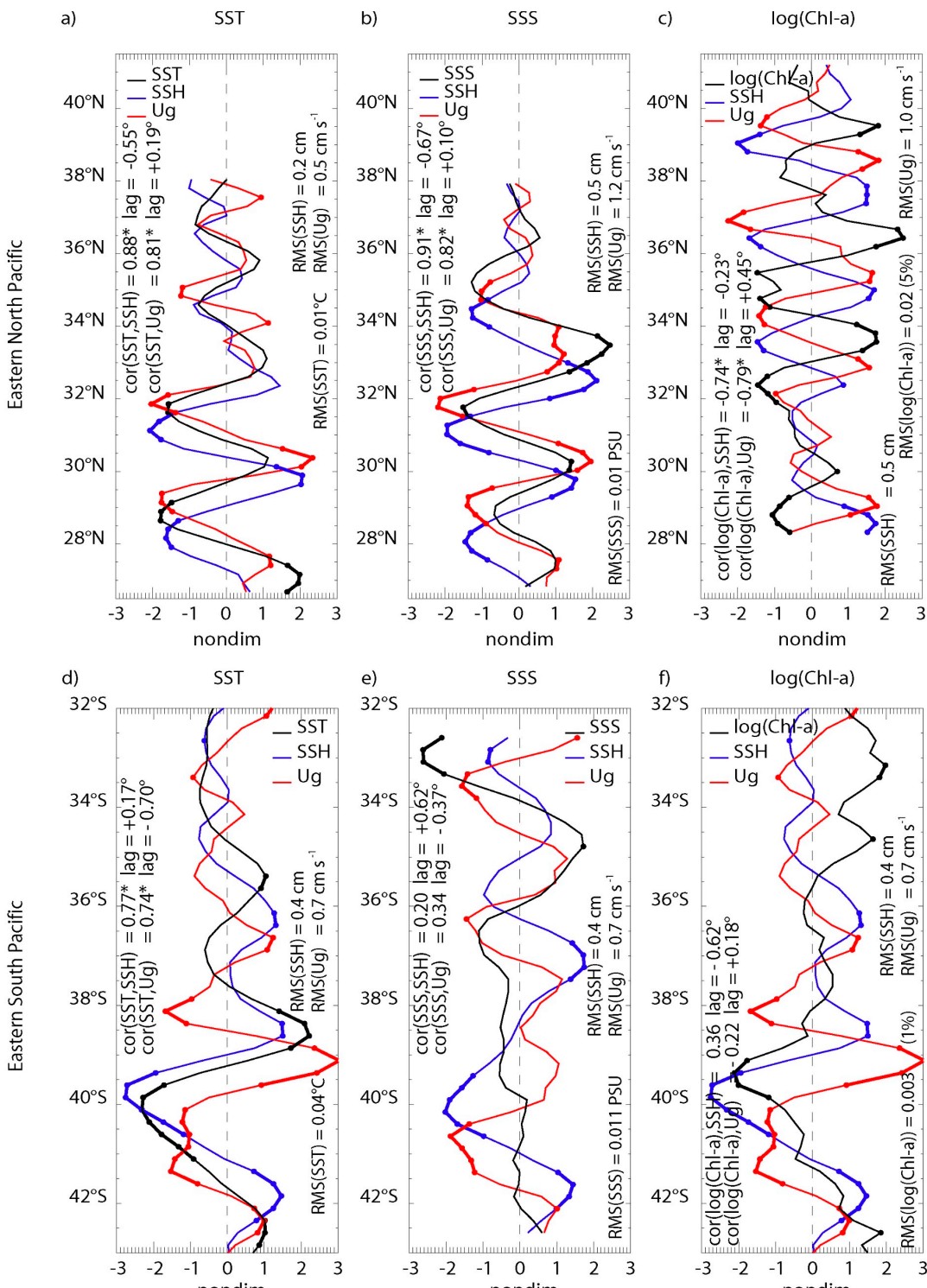

**Figure 6.** Cross-striation profiles of mean spatially high-pass filtered fields in the (**a–c**) ENP and (**d–f**) ESP, averaged quasi-zonally within the tilted solid boxes on Figures 2 and 3, and normalized by their standard deviation: black lines are for (**a,d**) AMSR-2 SST, (**b,e**) SMAP SSS, and (**c,f**) GlobColour log(Chl-a). The blue and red lines on all panels are for SSALTO/DUACS SSH and $U_g$, respectively. The dots joined by the thick lines indicate where each profile is statistically significant at the 5% level (see Appendix C). The maximum positive or negative lag-correlations of each tracer profile with both SSH and $U_g$ are indicated, together with their associated lag (positive lag is when the tracer field is shifted southward). The stars (*) indicate when the correlation is significant at the 5% level (see Tables S1 and S2). The standard deviation of each profile before normalization is also indicated, (**c,f**) together with that of the associated Chl-a profile expressed as a fraction of full Chl-a (%, see Figures 2e and 3e, Appendix A).

*3.2. ENP Tracer Advection*

In this section, we focus on time-averaged striated patterns in SST and SSS in the ENP that are consistent with the advection of large-scale water masses by the near-zonal jets, and perform a decomposition of the associated advection terms (Section 2.2). For both tracers, and as expected, tilted near-periodic zonally-elongated bands were found in the advection of the large-scale mean tracers by the mesoscale time-mean along-striation currents $-\overline{U_{aH}}\partial\overline{F_L}/\partial x_a$ (Figure 7a,c). SST advection was stronger near the eastern boundary (Figure 7a), consistently with both the swifter mesoscale zonal currents (Figure 2g) and the stronger background near-zonal SST gradients associated with coastal upwelling (Figure 2b). SSS advection was highest much further offshore, between ~140° W and ~130° W, and weaker closer to the coast (Figure 7c). This is likely the result of the structure of the background SSS gradient oriented mostly in the alongshore direction, which coincided with the cross-striation axis (Figure 2d). For both SST and SSS, the banded structures in $-\overline{U_{aH}}\partial\overline{F_L}/\partial x_a$ match those in $\overline{F_H}$ quite well (contours on Figure 7a,c), indicating that this term contributes to the generation of striated tracers (see Section 2.2).

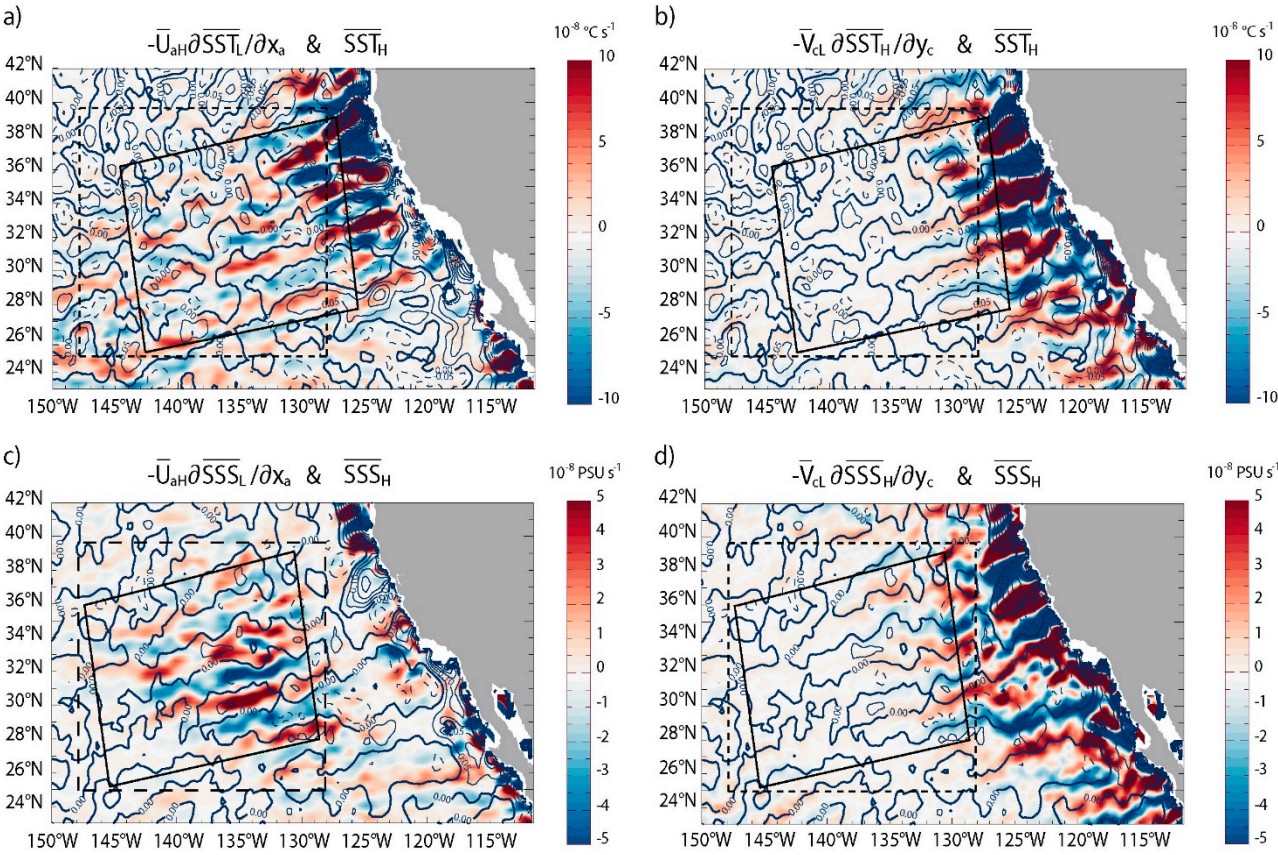

**Figure 7.** Advection of (**a,b**) SST and (**c,d**) SSS by striations in the ENP: (**a**) $-\overline{U_{aH}}\partial\overline{SST_L}/\partial x_a$; (**b**) $-\overline{V_{cL}}\partial\overline{SST_H}/\partial y_c$ ($10^{-8}$ °C s$^{-1}$); (**c**) $-\overline{U_{aH}}\partial\overline{SSS_L}/\partial x_a$; and (**d**) $-\overline{V_{cL}}\partial\overline{SSS_H}/\partial y_c$ ($10^{-8}$ PSU s$^{-1}$). The contours (solid/dashed for positive/negative values) are for (**a,b**) $\overline{SST_H}$ (CI = $10^{-2}$ °C) and (**c,d**) $\overline{SSS_H}$ (CI = $10^{-2}$ PSU). Solid and dashed boxes are as in Figure 2a,c, except that the solid boxes on the top panels are displaced 3 degrees eastward.

Similar bands are also found in the advection of mesoscale mean tracers by the large-scale mean cross-striation flow $-\overline{V_{cL}}\partial\overline{F_H}/\partial y_c$, except they are located near the continental boundary for both SST and SSS, and weaken sharply west of ~130° W, although they retain their banded structure through 135–140° W (Figure 7b,d). This term is banded because its mesoscale structure is defined by the cross-striation tracer gradients (Figure 2a,c). The large-scale surface flow is typical of subtropical gyres with equatorward (i.e., cross-striation) currents in the eastern branch and weaker velocities west of ~130° W

(Figures 1a and 2h), thus modulating the intensity of $-\overline{V_{cL}}\partial\overline{F_H}/\partial y_c$. Unlike the previous term, it appears to be in phase quadrature with the striated mesoscale SST/SSS (contours on Figure 7b,d). This is simply because $\overline{F_H}$ being near-periodic (Figure 6a,b), it is in phase quadrature with $\partial\overline{F_H}/\partial y_c$.

In an attempt to synthesize the information obtained from the 16 SST and SSS advection terms displayed in Equation (10), Figures 8 and 9 present the corresponding cross-striation profiles, as well as those of the mesoscale SST and SSS. The profiles were derived after averaging in the along-striation direction, as described earlier, except for the tilted box for the SST advection, which was shifted by three degrees eastward to better grasp the $-\overline{V_{cL}}\partial\overline{F_H}/\partial y_c$ term (Figure 7b). Figures 8 and 9 confirm that the aforementioned two terms are the dominant advection terms. In addition, the eddy terms (advection of the mesoscale tracer anomalies by the mesoscale anomalous along-striation and cross-striation currents, $-\overline{U_{aH}{}'\partial F_H{}'/\partial x_a}$ and $-\overline{V_{cH}{}'\partial F_H{}'/\partial y_c}$) also present mesoscale variations along the cross-striation axis, although with no obvious periodicity (Figures 8d and 9d). Importantly, these four mesoscale advection terms are not negligible compared to large-scale advection, especially for SSS (Figures 8a and 9a). Moreover, these terms are similar in magnitude for SST, while $-\overline{U_{aH}}\partial\overline{SSS_L}/\partial x_a$ is 2–3 times higher than other SSS advection terms. Such a result should, however, be interpreted with caution as it is likely sensitive to the location of the box used to compute the cross-striation profiles, due to spatial variations in the intensity of the different advection terms (Figure 7). The sensitivity to the choice of the averaging period is discussed in Section 4.

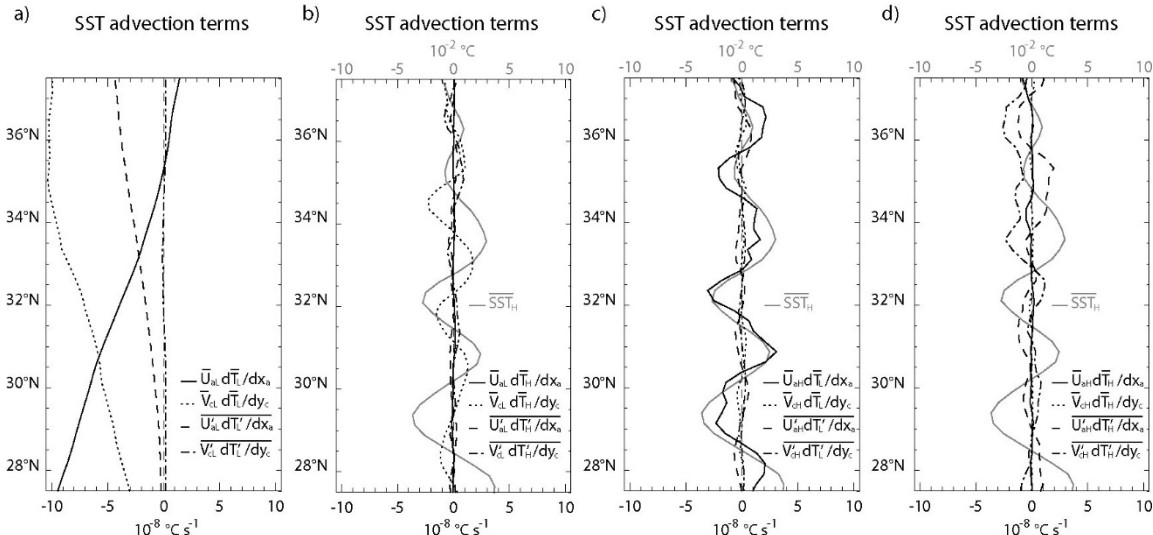

**Figure 8.** Cross-striation profiles (off the coast of California) of the SST advection terms (as listed in equation (10)), averaged quasi-zonally within the tilted solid box on Figure 7a,b. Each panel displays four terms (mean and eddy along-striation and cross-striation components) accounting for: (**a**) the advection of large-scale SST by the large-scale flow; (**b**) the advection of mesoscale SST by the large-scale flow; (**c**) the advection of large-scale SST by the mesoscale currents; and (**d**) the advection of mesoscale SST by the mesoscale currents. The profile of mean spatially high-pass filtered SST is also shown in panels (**b**–**d**).

The lag-correlation analysis of profile data confirms (1) the in-phase relationship between mesoscale physical tracers and $-\overline{U_{aH}}\partial\overline{F_L}/\partial x_a$ with high (~0.8), statistically significant correlations, and small lag values ~0.05–0.15° (Figures 8c and 9c, Table S3); and (2) the phase quadrature between $\overline{F_H}$ and $-\overline{V_{cL}}\partial\overline{F_H}/\partial y_c$ with significant ~0.8–0.9 negative correlations at lags ~0.8–0.9°, and the advection shifted equatorward (Figures 8b and 9b, Table S3). Considering both its intensity and alignment with $\overline{F_H}$, $-\overline{U_{aH}}\partial\overline{F_L}/\partial x_a$ therefore appears as the leading generation term for time-mean striated SST and SSS signals (see Section 2.2).

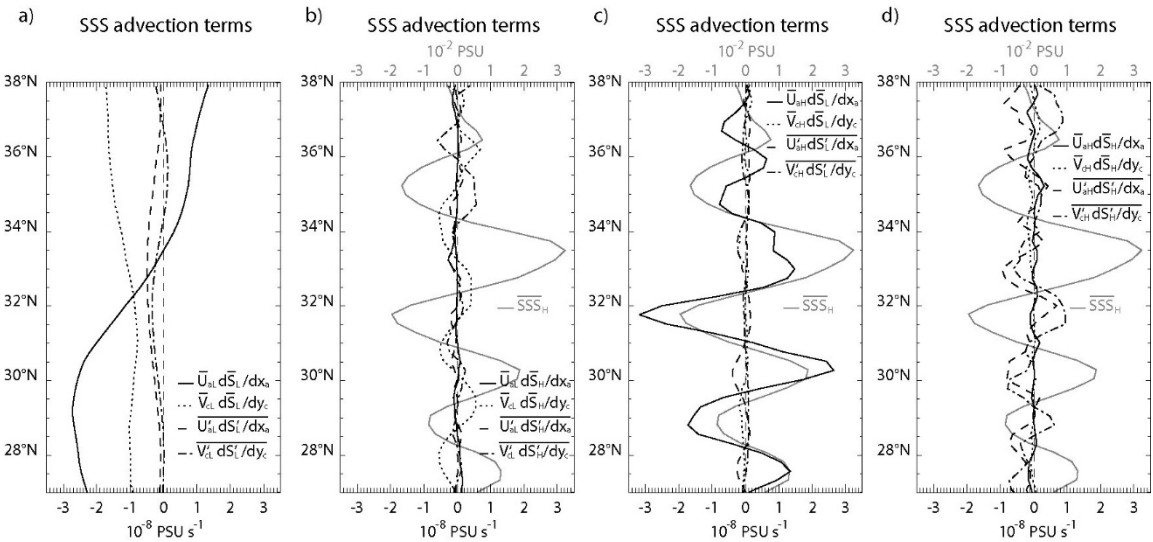

**Figure 9.** Same as Figure 8, except for the SSS advection terms and mean spatially high-pass filtered SSS, averaged quasi-zonally within the tilted solid box on Figure 7c,d.

Subsequently, we estimated the generation/dissipation time scale $t_0$ with the ratio of the standard deviations of $\overline{F_H}$ and $-\overline{U_{aH}}\partial \overline{F_L}/\partial x_a$ (Section 2.2). Estimates for SST (14 days) and SSS (12 days) were consistent with each other. They suggested that zonal advection by oceanic jets was a relatively fast process, which likely played a key role in the zonally anisotropic structuring of surface tracer fields at the mesoscale. Associated Rossby numbers *Ro* may be obtained from $1/t_0f$. In the ENP (27–38° N, Figures 8 and 9), $Ro$~0.9–1.2 $10^{-2}$ (1.1–1.5 $10^{-2}$) for SST (SSS). These values are close to those most frequently found for long-lived mesoscale eddies in the 20° N–60° N latitude band globally: 1–2 $10^{-2}$ [58]. These findings suggest that the time-averaged striations themselves may be related with mesoscale eddies, possibly organized in trains, and consistently with previous studies [14,16,57]. Artifacts of time-averaging random eddies [29] have indeed been proven to be inconsistent with a number of observed striation characteristics [13,16,19]. On the other hand, organized eddies suggest that striations may be the long-term signature of the dynamics underlying the organization of the eddy field, such as nonlinear beta-plumes [15,16,38] or radiating instability [31]. The possibility that striations themselves may be dynamic structures interacting with eddies and resulting in such eddy alignments has also been proposed [8,16]. These eddy trains may, therefore, be major drivers of the striated tracer pattern formation, through their time-averaged signature as striations of the zonal current.

## 4. Discussion

The mesoscale SST and SSS budgets depicted above are not closed. Other terms in the tracer balance (Equation (1)) that were not explicitly included in this study, such as horizontal/vertical mixing and vertical advection, may also be important, some of them likely participating to the dissipation of striated tracer features. For instance, the vertical advection of nutrients into or out of the euphotic layer may be particularly relevant for surface chlorophyll, as has been shown for mesoscale eddies [59]. Future modelling studies at eddy-resolving resolution (typically 0.1° and higher) may help to address this limitation.

In light of our results, the processes responsible for the generation and persistence of striated patterns in SST and SSS may be conceptualized as follows. Mean quasi-zonal jets in the ENP advect the large-scale tracer gradients extending between the coast and the subtropical gyre, leading to tracer field deformation (Figure 10a,b; see also [37], their Figure 13a). The resulting wavy pattern in the initially near-meridional (i.e., cross-jet) tracer isolines generates near-zonal frontal anomalies that appear as a striated signal in the

time-mean mesoscale field. These features are, however, partly embedded in a large-scale equatorward flow typical of subtropical gyres and EBUS. Considering the periodicity of the time-averaged mesoscale tracer signals along the cross-striation axis, such a large-scale current then acts to advect mesoscale warm/salty anomalies from the north, into the tracer front that separates them from the next near-zonal band of cooler/fresher waters, to the south, and, similarly, cool/fresh anomalies into the front delineating a band of warmer/saltier waters (Figure 10c, left). However, cross-striation advection alone does not explain the banded tracer pattern, given its shorter near-zonal extent and the phase lag with the tracer field, suggesting that this secondary mechanism may be slower and therefore less efficient. Other processes, such as horizontal eddy advection (Figure 10c, left), mixing, and vertical advection (Figure 10c, right), likely contribute to the dissipation (and possibly to the generation) of mesoscale SST/SSS patterns as well, allowing for equilibrium and for the maintenance of the footprint of striations in SST and SSS.

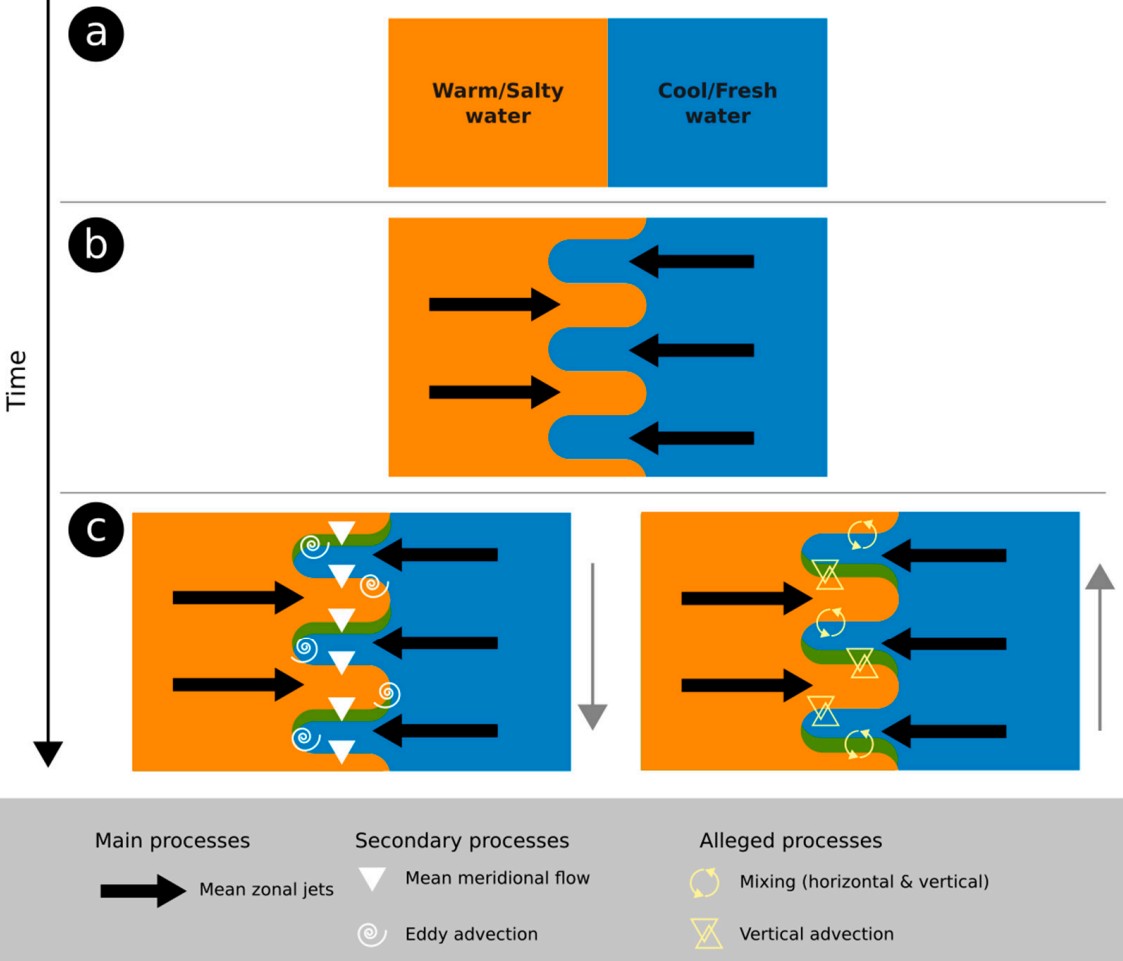

**Figure 10.** Conceptual model for the time-averaged impact of striations on SST and SSS in the ENP. (**a**) Initial state: large-scale zonal gradient between the cooler and fresher surface waters of the California upwelling system and the warmer and saltier waters of the North Pacific subtropical gyre. (**b**) Main process: multiple time-mean zonal jets advect the background SST and SSS gradients, rapidly generating striated tracer anomalies in the form of alternating mesoscale zonal fronts. (**c**) Secondary and alleged processes: (left) the large-scale meridional flow acts to displace the mesoscale tracer pattern equatorward (grey arrow directed southward and green areas to the south of the warm/salty tongues) by advecting warm/salty and cool/fresh waters across the fronts on their southern flanks, with additional contributions from eddy advection; (right) vertical advection, horizontal and vertical mixing likely balance the aforementioned processes, through the compensating poleward shift of tracer anomalies (grey arrow directed northward and green areas to the north of the warm/salty tongues) and the dissipation of the striated SST/SSS anomalies originally generated through zonal advection.

Unlike the expression of mean striations in SST and SSS, the one in Chl-a appears $\pi$ out of phase with the striated SSH signal in the ENP. Cyclonic mesoscale eddies are thought to contribute to the offshore nutrient and plankton export in EBUS by trapping coastal water during their formation, unlike anticyclonic eddies generated beyond the upwelling front in more oligotrophic environments [60,61]. The observed relationship between the bands in Chl-a and SSH is then consistent with such a hypothesis, given that time-mean striations manifest themselves in snapshots as eddy trains [14,16,57]. However, upwelled trapped water is also cooler compared to background open-ocean conditions, which should lead to an in-phase SST–SSH relation, rather than the phase quadrature evidenced here. Such a discrepancy may result from the different locations for the studied physical and biological tracers in the offshore and coastal transition regions, respectively, where different dynamics may be involved.

To verify this hypothesis, cross-striation profiles of SST and SSS are computed in the same region as Chl-a (tilted box on Figure 2e) and shown in Figure 11. As for the offshore region (Figure 6a), several peaks of the mesoscale SST profile are statistically significant at the 5% level, particularly near 35–39° N and 28–29° N. SST is highly (~0.8) and significantly correlated with both SSH and $U_g$ (Table S4), with phase lags ~−0.5° and ~+0.25° latitude, respectively. The smaller lag with $U_g$ suggests that eddy trapping may not be the dominant driver for the striated mean SST pattern in this region. Indeed, the correlation at zero lag is significant with $U_g$ only, and higher (0.69) compared to SSH (0.38). Nevertheless, the examination of Figure 11a reveals that SST is more aligned with $U_g$ north of ~37° N and with SSH to the south. Thus, these results suggest that both eddy trapping and advection likely contribute to the striated tracer patterns in the coastal transition region. Note that SSS is poorly correlated with both SSH and $U_g$ (Table S4), and its periodicity is not as marked as for other variables with fewer robust peaks (Figure 11b), which may have to do with the mostly alongshore background SSS gradient (Figure 2d).

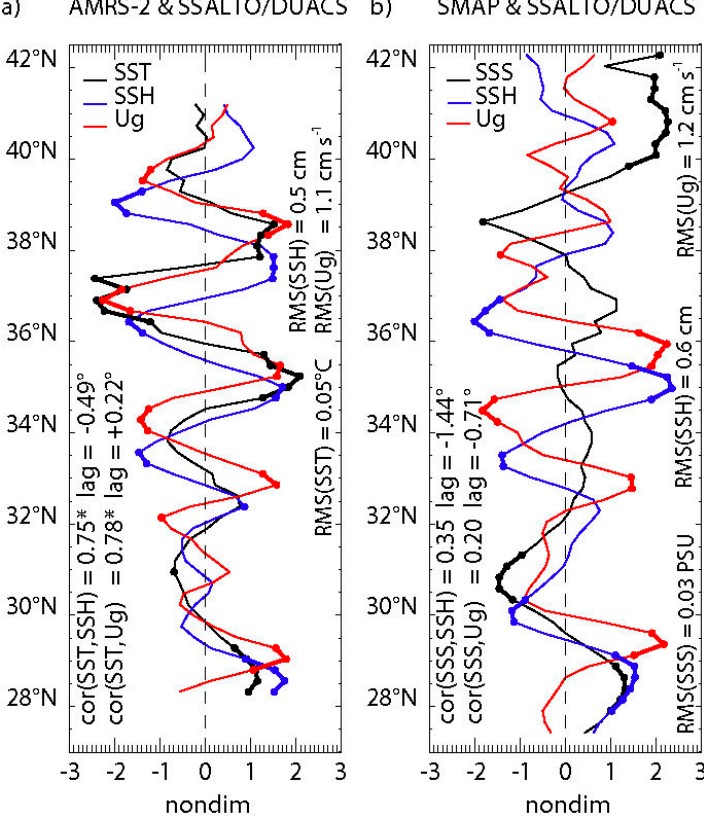

**Figure 11.** Same as Figure 6a,b, except averaged within the tilted solid box on Figure 2e. The stars (*) indicate when the correlation is significant at the 5% level (see Table S4).

In contrast, the average footprint of striations in the ESP is only evident in the SST field. The phase quadrature between the latter and geostrophic velocity in a region of a marked meridional SST gradient suggests a potential role of inhomogeneous isopycnal mixing [36,37].

Except for Chl-a in the ENP, the time-mean striated tracer signals reported here are relatively weak (yet robust). Nevertheless, the associated mesoscale meridional SST (SSS) gradients are typically $\pm$ 10–20% ($\pm$20–50%) of the large-scale gradient and locally over $\pm$ 30% (near $\pm$ 100%) (Figure 12), implying a significant modulation of zonal tracer fronts by striated currents. Moreover, instantaneous tracer anomalies associated with eddy trains are likely much stronger than time-mean signals as one might expect from the SST/SSS signatures of mesoscale eddies [14,62]. A composite analysis of those eddies organized in striations may be necessary to extract these transient signals and their contribution to SST/SSS variability in EBUS.

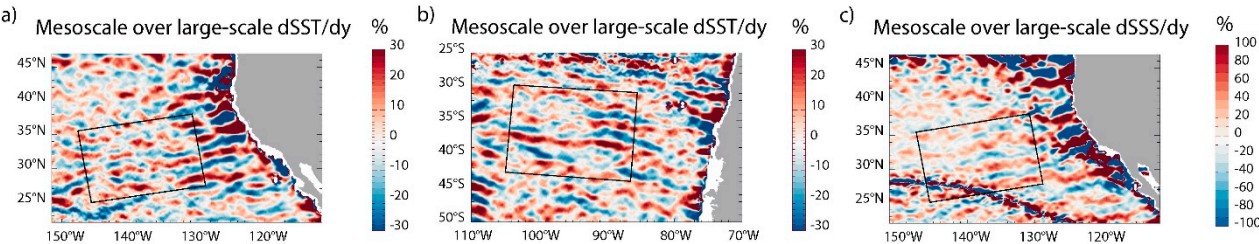

**Figure 12.** Ratio of spatially high-pass filtered to spatially low-pass filtered time-averaged meridional gradient of (**a**,**b**) AMSR-2 SST and (**c**) SMAP SSS in the (**a**,**c**) ENP and (**b**) ESP. The tilted boxes on (**a**,**c**) and (**b**) are the same as in Figure 2a,c and Figure 3a, respectively.

As stated in the introduction, the stationarity of striations over long time scales is an open question [5,6,12–27], although different dynamical structures with distinct kinematics may be improperly grouped under the same generic term "striations". The present study focuses on the average effects of striations on surface tracers over multi-year periods, as a first approach, while transient effects remain out of our scope. Nevertheless, it can be useful to assess the sensitivity of the time-mean quasi-zonal jets to the exact averaging period, in order to anticipate the possible associated sensitivity of surface water properties. Figure 13 represents the Hovmoller diagrams of the time-averaged mesoscale $U_g$ cross-striation profiles previously shown in Figure 6a,d, but considering all the possible ~6.5 year averaging periods within the ~27.5-year available altimetric records. Several remarks may be inferred from these plots. First, striations defined in such a way exhibit significant sensitivity to the averaging multi-year period in the two study regions. This is particularly true in the ENP, where the low-frequency modulation of long-term mean zonal jets is prominent compared to the somewhat steadier, yet variable, ESP striations. Second, and despite such marked temporal variability, striations do show mostly stationary behavior over particular (subtropical) latitude ranges and time periods, namely from 30° N to 38° N during the first half of the record and from 33° S to 39° S during the second half. This is consistent with the quasi-stationary nature of ENP/ESP striations shown in some previous studies [5,8,13,16]. Third, meridional propagation seemingly occurs dominantly equatorward, consistently with several observational and model studies pointing out such kinematics over the latitude ranges of our study regions [6,19,26], or slightly closer to the tropics [18,22]. Yet, Figure 13 also suggests that, in addition to slow, steady drift, such propagation can occasionally take place in the form of sudden latitudinal shifts limiting periods of relatively steady behavior, particularly in the ENP, and consistently with similar observations made by Ref. [6] and Ref. [16] off the coasts of California and Chile, respectively. These drifts and shifts are, however, not unequivocal evidence that striations are transient rather than quasi-stationary: for instance, changes in striation angles resulting from, e.g., changes in the meridional flow intensity [5], may also explain such

variability (Figure 13 considers a constant striation angle corresponding to the ~6.5-year study period). Last, the particular timing of the 6.5 year period considered in this study is not necessarily representative of the differences in $U_g$ between the ENP and ESP over longer time scales. For instance, the intensity of the eastward jet near 39° S is unprecedented, while ENP striations tend to be weaker compared to e.g., 10 years earlier. Considering the previously demonstrated importance of zonal advection by the mean striations in this region, this finding may have strong implications in terms of the intensities of striated SST/SSS patterns and of the associated advection terms, as quantified in Section 3.2. While swifter zonal jets in the 1990s and 2000s may be associated with larger tracer anomalies compared to our results, future studies should concentrate on a more detailed analysis of the temporal variability of striations (e.g., using empirical orthogonal functions [23]) and their impacts on ocean properties in order to expand our understanding of this topic.

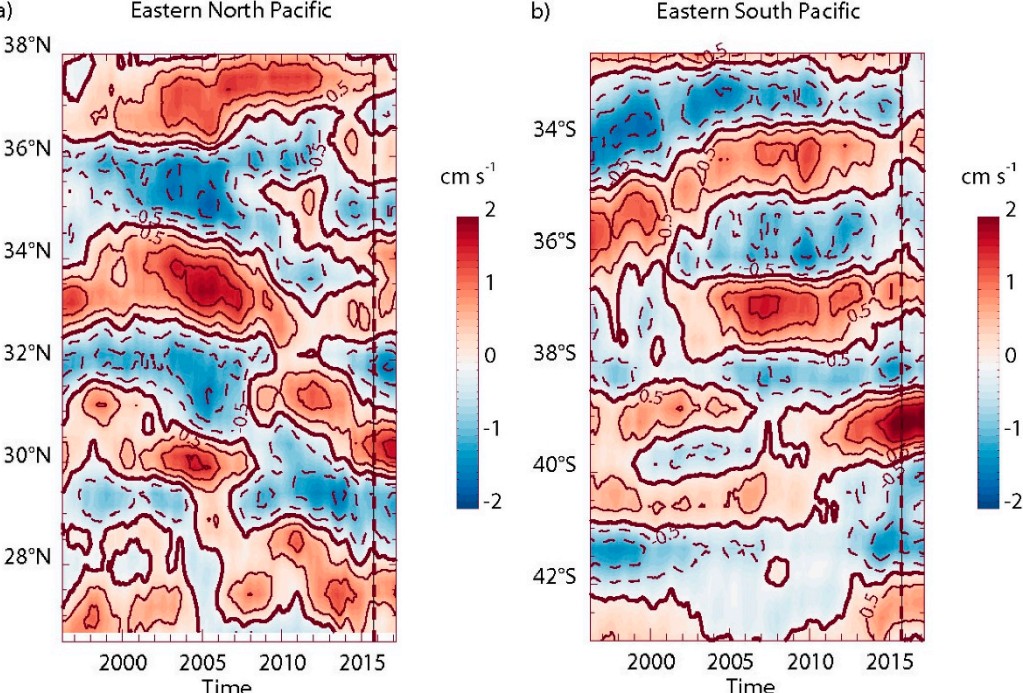

**Figure 13.** Time evolution of spatially high-pass filtered SSALTO/DUACS $U_g$ (cm s$^{-1}$) over the full available altimetric record (1 January 1993–3 June 2020) in the (**a**) ENP and (**b**) ESP, averaged quasi-zonally within the offshore tilted solid boxes in Figures 2g and 3g. A moving average of width equal to that of the study period (2 July 2012–31 December 2018 ~6.5 years) was applied to the time series of cross-striation profiles. The dashed vertical line near the end of the time series marks the data over the study period on both panels, corresponding to the $U_g$ profiles shown in Figure 6a,d (see also, Figures S3c and S6c).

Overall and despite their similarities, the Californian and Chilean EBUS feature striated patterns of surface tracers with quite distinct characteristics. This is attributed mostly to the location of quasi-zonal jets, either within the large-scale near-zonal tracer gradients and meridional flow (ENP) or much farther west (ESP). These variations among regions, time periods, and tracers make it difficult to generalize our results and apply them to other EBUS regions, such as the Benguela and Canary current systems in the Atlantic. Diverse theories for the existence of striated currents [15,16,28–37] suggest the possibility of different generation mechanisms in different regions. Combined with the variety of water masses in the global ocean and their spatial properties, particularly the strength and direction of background tracer gradients, such diversity suggests large regional variations in the existence and characteristics of striated tracer fields.

## 5. Conclusions

For the first time, multi-year satellite records from various sensors were used to characterize the time-averaged effects of striations on SST, SSS, and Chl-a in the subtropical ENP and ESP EBUS. The results vary significantly among the two regions and three surface tracers. In the ENP, while the expression of striations in SST and SSS coincide with that in zonal current, suggesting a dominant role of advection, the expression of striations in Chl-a is collocated with that in SSH, possibly involving water mass trapping by mesoscale eddy trains. In the ESP, striated patterns were found in SST but not in SSS nor in Chl-a, except for some meridionally-alternating Chl-a anomalies in the coastal transition zone without clear connections to quasi-zonal jets. Moreover, striated SST signals are highly correlated with SSH, rather than with zonal currents. Unlike the ENP, ESP striations are located far offshore, outside the area of strong background zonal gradients of ocean properties, which explains the weak effects of zonal advection. Therefore, striations probably contribute to coast/open-ocean exchanges in the ENP, but this is unlikely for the ESP. Our findings expand the current knowledge of these exchanges and advocate for further research to understand the complex influences of striations on ocean properties and their variations between different regions and periods of time.

The decomposition of SST and SSS advection in the ENP into along/cross-striation, mean/eddy, and mesoscale/large-scale parts identifies a dominant contribution from the advection of large-scale mean tracers by the mean quasi-zonal jets, which acts as a fast generation process for striated tracer signals, and secondary contributions from the advection of mesoscale mean tracers by the large-scale mean meridional flow and from eddy advection, although mixing and vertical advection likely contribute as well to the generation and dissipation of striated tracers. Assessing these latter terms of the surface tracer balance will require the analysis of numerical ocean models and is left for future studies, which should also investigate the associated sensitivity to the chosen study period. Based on the presently available satellite data, the persistence of the hydrographic signature of striations is suggested to result from an interaction between mesoscale (quasi-zonal jets, fronts, and eddies) and large-scale features (tracer gradients and meridional flow). The processes leading to the alignment of the SSH signature of striations with Chl-a in the ENP and SST in the ESP are not confirmed and require future investigation.

**Supplementary Materials:** The following are available online at https://www.mdpi.com/article/10.3390/fluids6120455/s1, Figure S1: Striation expressions in hydrographical tracers in the two study regions from alternate data; Figure S2: Anisotropic ratio for hydrographical tracers in the two study regions from alternate data; Figure S3: Cross-striation profiles of spatially high-pass filtered mean SST, SSH, and $U_g$ in the ENP; Figure S4: Same as Figure S3 except for SSS; Figure S5: Same as Figure S3 except for log(Chl-a); Figure S6: Same as Figure S3 except for the ESP; Figure S7: Same as Figure S4 except for the ESP; Figure S8: Same as Figure S5 except for the ESP; Figure S9: Same as Figure S3 except for the ENP coastal transition zone; Figure S10: Same as Figure S4 except for the ENP coastal transition zone; Table S1: Results of the Monte Carlo analysis for the correlation coefficients shown on Figure 6a–c; Table S2: Results of the Monte Carlo analysis for the correlation coefficients shown on Figure 6d–f; Table S3: Results of the Monte Carlo analysis for the correlation coefficients between cross-striation profiles of $\overline{F_H}$ and $-\overline{U_{aH}}\partial \overline{F_L}/\partial x_a$ or $-\overline{V_{cL}}\partial \overline{F_H}/\partial y_c$; and Table S4: Results of the Monte Carlo analysis for the correlation coefficients shown on Figure 11.

**Author Contributions:** Conceptualization, A.B., P.-A.A., N.M. and S.C.; methodology, A.B., P.-A.A., K.G. and N.M.; software, A.B., P.-A.A. and K.G.; validation, A.B. and P.-A.A.; formal analysis, A.B. and K.G.; investigation, A.B. and P.-A.A.; resources, A.B.; data curation, A.B.; writing—original draft preparation, A.B.; writing—review and editing, A.B., P.-A.A., N.M. and S.C.; visualization, A.B. and P.-A.A.; supervision, A.B.; project administration, A.B.; funding acquisition, A.B., P.-A.A. and N.M. All authors have read and agreed to the published version of the manuscript.

**Funding:** This research was funded by the Office of Naval Research Global, Grant No. N62909-16-1-2228. N.M. is partly supported by the National Aeronautics and Space Administration through Grants NNX17AH43G and 80NSSC20K0891.

**Institutional Review Board Statement:** Not applicable.

**Informed Consent Statement:** Not applicable.

**Data Availability Statement:** The SSALTO/DUACS altimeter ADT and derived variables were produced and distributed by the Copernicus Climate Change Service (C3S) (https://cds.climate.copernicus.eu/cdsapp#!/dataset/satellite-sea-level-global?tab=overview, last accessed on 14 October 2021). This study has been conducted using E.U. Copernicus Marine Service Information (CMEMS). The GlobCurrent data are available at https://resources.marine.copernicus.eu/?option=com_csw&view=details&product_id=MULTIOBS_GLO_PHY_REP_015_004, accessed on 14 October 2021. This study used the SCUD [46] surface velocities provided by APDRC/IPRC. AMSR-2 data are produced by Remote Sensing Systems and were sponsored by the NASA AMSR-E Science Team and the NASA Earth Science MEaSUREs Program. Data are available at www.remss.com/missions/amsr/, accessed on 14 October 2021. The OSTIA data were provided by GHRSST, Met Office, and CMEMS. The L3_DEBIAS_LOCEAN_v4 Sea Surface Salinity maps have been produced by the LOCEAN/IPSL (UMR CNRS/UPMC/IRD/MNHN) laboratory and ACRI-st company that participate in the Ocean Salinity Expertise Center (CECOS) of Centre Aval de Traitement des Donnees SMOS (CATDS). This product is distributed by the CECOS of the CNES-IFREMER CATDS, at IFREMER, Plouzané (France). Satellite data of sea surface chlorophyll are from the GlobColour product available at https://hermes.acri.fr/, accessed on 14 October 2021.

**Acknowledgments:** We thank Ru Chen and two anonymous reviewers whose constructive comments were key for the improvement of the original submission. Sébastien Hervé is thanked for his assistance with the design of the conceptual sketch (Figure 10). The APC was funded by LOPS (Ifremer).

**Conflicts of Interest:** The authors declare no conflict of interest. The funders had no role in the design of the study; in the collection, analyses, or interpretation of data; in the writing of the manuscript; or in the decision to publish the results.

## Appendix A

A logarithmic transformation is applied to time averages of monthly Chl-a data as stated in Section 2.2. Let $A = \log(\text{Chl-a})$. A is high-pass filtered as described in Section 2.2. By construction, mesoscale A is then the difference between A and large-scale A, which may be viewed as an anomaly relative to the large-scale component:

$$A_H = A - A_L \tag{A1}$$

where $H$ and $L$ subscripts refer to mesoscale and large-scale components, respectively.

For 2D visualization purposes, it is useful to represent Chl-a expressed in mg m$^{-3}$ with a logarithmic color scale (Figures 2f and 3f). This is conducted by plotting A with a linear color scale, and indicating the associated Chl-a values. For consistency, it is desirable to represent $A_H$ in such a way that it may be compared quantitatively with full (unfiltered) Chl-a. Raising 10 to the power of Equation (A1) after moving $A_L$ to the left hand side yields:

$$\text{Chla} = \gamma 10^{A_L} \tag{A2}$$

where $\gamma = 10^{A_H}$. Permutating low-pass filtering and logarithmic transformation does not make much difference on large-scale Chl-a outside a narrow coastal band in both the ENP and ESP (not shown). Therefore:

$$\text{Chla} \approx \gamma \text{Chla}_L \tag{A3}$$

Using scale separation as in (A1), but applied in the original Chl-a space, it can be shown that:

$$\text{Chla}_H \approx \text{Chla}(\gamma - 1)/\gamma \tag{A4}$$

The ratio between mesoscale and full Chl-a may thus be approximated with the nondimensional coefficient $(\gamma - 1)/\gamma$, which can be readily derived from $A_H$. A linear color scale is then used for $A_H$, which we made vary from $-0.05$ to $0.05$ in units of the

logarithmic space. The associated $(\gamma - 1)/\gamma$ values then range approximately from $-0.11$ to 0.12 (Figures 2e and 3e).

In contrast, the cross-striation profiles of mesoscale log(Chl-a) do not require such inverse transformation because of the normalization by their standard deviation (Figure 6c,f). The latter may be expressed either in units of the logarithmic space (i.e., of $A_H$) or as a ratio between mesoscale and full Chl-a (i.e., $(\gamma - 1)/\gamma$) according to the aforementioned derivation.

### Appendix B

As a complement to visual inspection, an objective approach is used to identify the regions where striated patterns in mesoscale mean velocity and tracer fields are most pronounced. Similarly to [30,55], anisotropy of the mesoscale mean geostrophic velocities is estimated with an anisotropic ratio, defined as:

$$\mathrm{M} = (\overline{u}_H{}^2 - \overline{v}_H{}^2)/(\overline{u}_H{}^2 + \overline{v}_H{}^2) \tag{A5}$$

where $\overline{u}_H$ and $\overline{v}_H$ refer to the high-pass filtered time-averaged zonal and meridional geostrophic velocities, respectively. $M$ is bounded by $-1$ and 1 (purely meridional and zonal mesoscale time-mean velocity, respectively) and equal to 0 when zonal and meridional components have equal magnitudes. This ratio is expected to be markedly positive in the areas where striations have been visually identified in maps of $\overline{u}_H$ as detailed in Sections 2.2 and 3.1 (Figures 2g and 3g).

Considering geostrophy, $M$ may be re-written in terms of SSH as:

$$M_{\overline{\mathrm{SSH}}_H} = \left[\left(\partial\overline{\mathrm{SSH}}_H/\partial y\right)^2 - \left(\partial\overline{\mathrm{SSH}}_H/\partial x\right)^2\right] / \left[\left(\partial\overline{\mathrm{SSH}}_H/\partial y\right)^2 + \left(\partial\overline{\mathrm{SSH}}_H/\partial x\right)^2\right] \tag{A6}$$

where $\overline{\mathrm{SSH}}_H$ is the high-pass filtered time-averaged SSH; and $x$ and $y$ are zonal and meridional coordinates, respectively. An analogy between scalar SSH and tracer fields may be used to define similar ratios for tracers as:

$$M_{\overline{F}_H} = \left[\left(\partial\overline{F}_H/\partial y\right)^2 - \left(\partial\overline{F}_H/\partial x\right)^2\right] / \left[\left(\partial\overline{F}_H/\partial y\right)^2 + \left(\partial\overline{F}_H/\partial x\right)^2\right] \tag{A7}$$

where $\overline{F}_H$ is the high-pass filtered time-averaged surface tracer field.

The application to SST and SSS is straightforward. For Chl-a, because of considerations associated with log-transformation, the Equation (A7) needs to be applied to the nondimensional fields shown on Figures 2e and 3e, i.e., to $(\gamma - 1)/\gamma$, where $\gamma = 10^{A_H}$ and A = log(Chl-a) (see Appendix A). Regardless of the tracer (or velocity) field under consideration, local values of the anisotropic ratio are somewhat noisy because of small-scale variability such as zonal variations in the banded patterns or zero-crossings along the cross-striation axis, for instance. To provide meaningful results, the mesoscale component of $M$ was removed with spatial low-pass filtering using a 2D Hanning window of $4°$ half-width. This allows to focus on regional-scale variations in $M$ and therefore to identify broad regions of marked zonal anisotropy.

Figures A1 and A2 present the results of this procedure for the ENP and ESP, respectively. As expected, the mesoscale time-mean currents are clearly zonally anisotropic across most of the two study regions, including the areas that were previously selected for the FFT analysis (solid and dashed rectangles on Figures A1d and A2d). In contrast, tracer fields exhibit both regions of zonal and meridional anisotropy.

In the ENP, the most prominent areas of positive $M$ values are well captured by the FFT subdomains for SSS and Chl-a (Figure A1b,c). The FFT subdomain for SST also grasps the region of zonal anisotropy to a large extent, although it is missing a part in the coastal transition zone located further east (Figure A1a). Such choice was, however, necessary because more energetic signals in this area (Figure 2a) would otherwise introduce undesired spatial biases in the FFT analysis. Unlike $M$, the subjective visual approach has

the advantage of discriminating regions of strong and weak magnitude of striated features, in addition to simplicity.

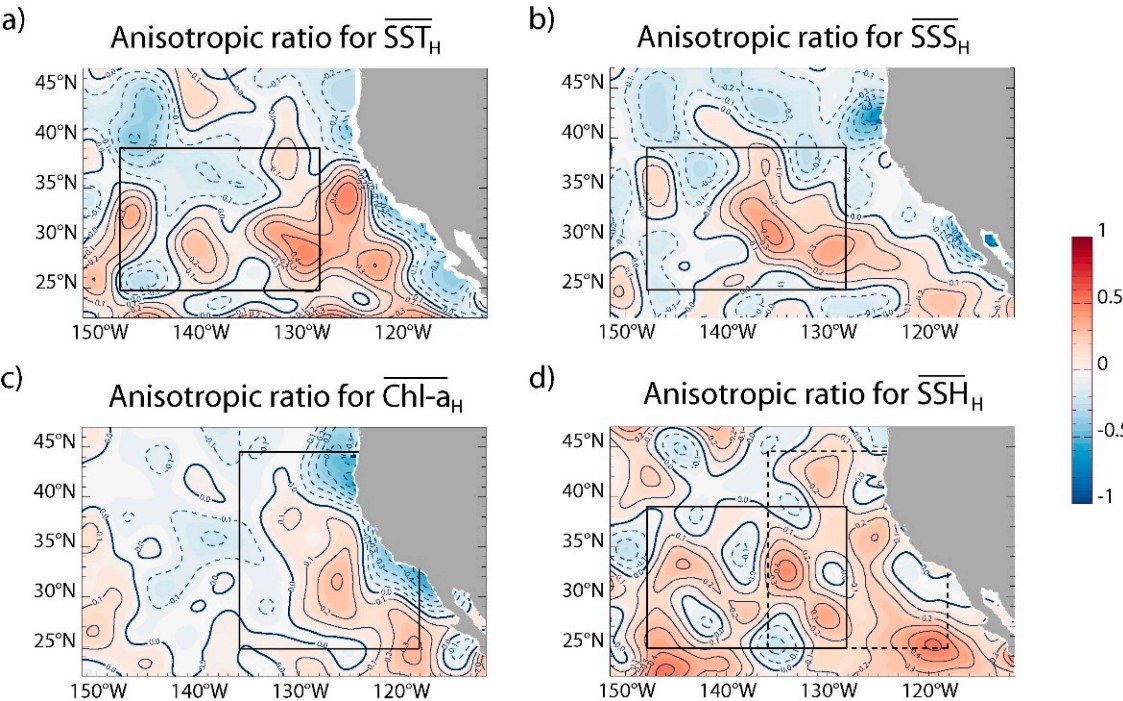

**Figure A1.** Anisotropic ratio for the spatially high-pass filtered time-averaged (**a**) AMSR-2 SST, (**b**) SMAP SSS, (**c**) GlobColour Chl-a, and (**d**) SSALTO/DUACS SSH in the ENP. The solid box on (**a,b,d**) corresponds to the dashed box on Figure 2a,c. The solid box on (**c**) and dashed box on (**d**) correspond to the black dashed box on Figure 2e.

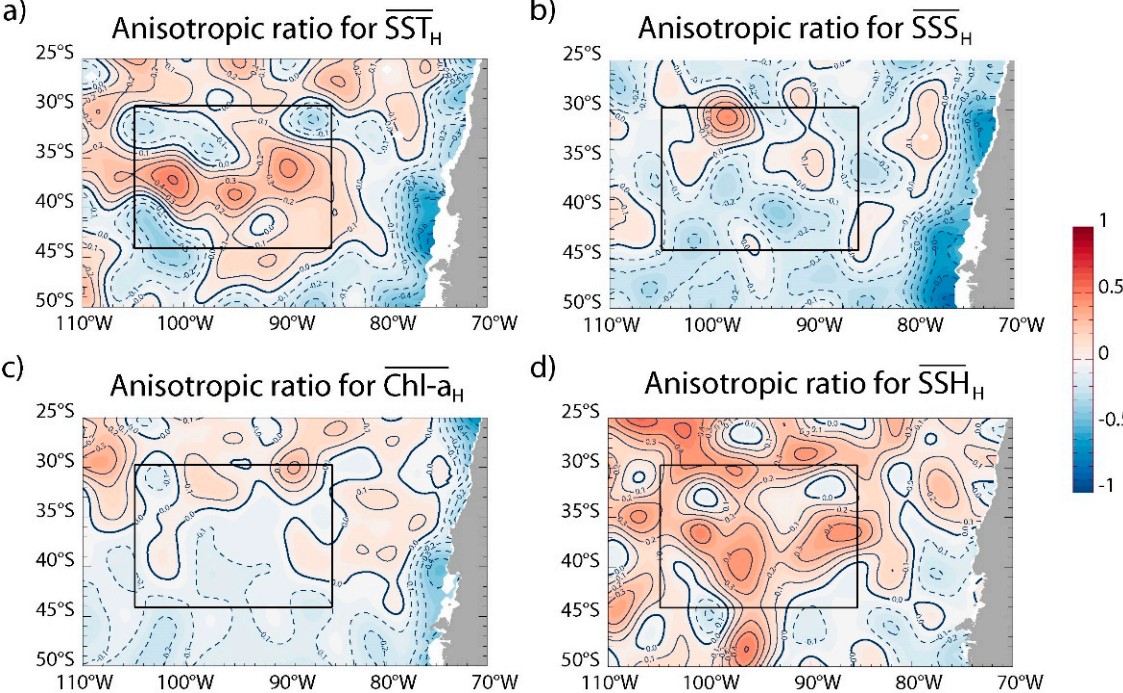

**Figure A2.** Same as Figure A1, except for the ESP. The solid box on all the panels corresponds to the dashed box on Figure 3a,c,e.

In the ESP, only the SST field presents clear zonally anisotropic signals in the common subdomain that was chosen for all the variables (Figure A2a). Furthermore, the regional distributions of *M* for SST and geostrophic velocities agree remarkably well, despite more pronounced meridional anisotropy for the former in specific regions including the coastal band (Figure A2a,d). In contrast, the mesoscale Chl-a field does not exhibit a clear tendency for either zonal or meridional anisotropy in the FFT subdomain, and is only weakly zonally anisotropic to the north and east (Figure A2c). SSS is mostly meridionally anisotropic across the study region, except for some isolated patches of positive *M* values (Figure A2b).

Overall, these results support the visual identification of areas with striated patterns, as inferred from Figures 2 and 3. The few discrepancies, particularly in the ESP, are not considered detrimental for subsequent analyses because the present objective approach could not identify any obvious alternate choices of regions with striated signals common to tracer fields and geostrophic velocity data.

**Appendix C**

Assuming that nominal errors of satellite sensors are normally distributed, confidence intervals can be obtained from the standard error of the mesoscale time-mean SST/SSS fields, which may be estimated as:

$$\mathrm{SE}_{F_H}(x,y) = \sigma_{F_H}(x,y)/\sqrt{n} \tag{A8}$$

where *F* is SST or SSS; subscript *H* refers to the mesoscale component; $\sigma_{F_H}$ is the standard deviation of mesoscale *F* time series; *n* is the number of independent observations of mesoscale *F*; and *x* and *y* are zonal and meridional coordinates, respectively.

In the regions where the time-mean hydrographic signature of striations is observed (tilted boxes on Figure 2a,c and Figure 3a), $\sigma_{\mathrm{SST}_H}$ is of order 0.25 °C and 0.35 °C in the ENP and ESP, respectively, whereas $\sigma_{\mathrm{SSS}_H}$ is of order 0.15 PSU in the ENP (Figure A3). Note that no striated SSS signals were found in the ESP (see Section 3).

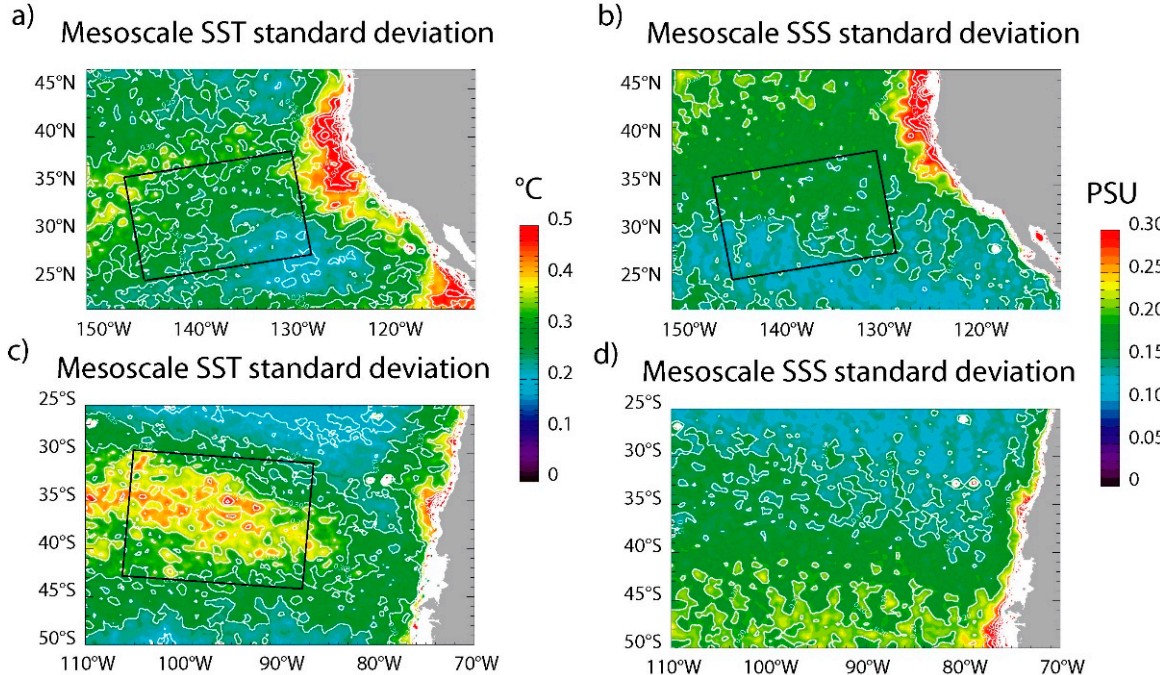

**Figure A3.** Standard deviation of spatially high-pass filtered (**a,c**) AMSR-2 SST (°C) and (**b,d**) SMAP SSS (PSU) in the (**a,b**) ENP and (**c,d**) ESP. The tilted boxes on (**a–c**) are as in Figure 2a,c and Figure 3a, respectively.

Moreover, striations are associated with the time-averaged signature of westward mesoscale eddy propagation [15,16]. A typical time scale $T$ separating two consecutive independent observations may thus be defined as the time spent by an individual eddy passing through a fixed location. Under the Gaussian eddy approximation, such time corresponds to eddy diameter $D$ divided by eddy zonal translation velocity $c$, which may be estimated from [58] (see their Figure 12 and Figure 22). According to their results, $c$ is typically 2 cm s$^{-1}$ in the 30–45° S latitude band (ESP) and 2.5 cm s$^{-1}$ in the 25–40° N band (ENP), while $D$ is roughly equal to 150 km in both regions. $T = D/c$ is then ~70 days and ~90 days in the ENP and ESP, respectively. Finally, $n$ is obtained by dividing the period of record (2374 days and 1368 days for SST and SSS, respectively) by $T$ and retaining the integer part of the result. This yields 33 and 26 independent SST observations in the ENP and ESP, respectively, and 19 independent SSS observations in the ENP. The application of Equation (A8) to the aforementioned estimates of $\sigma_{SST_H}$, $\sigma_{SSS_H}$, and $n$ then yields standard errors of SE$_{SST_H}$~0.04°C and 0.07°C in the ENP and ESP, respectively, and SE$_{SSS_H}$~0.03 PSU in the ENP.

90% confidence intervals around mesoscale time-mean $F$ values may then be defined as $\pm\,1.64$ SE$_{F_H}$ under the Gaussian hypothesis, which translates into $\pm0.07°$ C and $\pm0.11$ °C in the ENP and ESP, respectively, and $\pm0.06$ PSU in the ENP. Note that since the normal distribution is two-tailed, the probability that the true mesoscale time-mean $F$ is lower than the lower limit of the 90% confidence interval associated with positive anomalies shown on Figure 2a,c and Figure 3a is only 5%, and similarly for the upper limit of the 90% confidence interval associated with negative anomalies. Any zero-crossings of 90% confidence intervals may then be interpreted as the data not being significantly different from zero at the 5% level, and vice-versa.

A similar approach is followed to obtain 90% confidence intervals for the mesoscale temporally and near-zonally averaged profiles of all variables shown on Figures 6 and 11. The associated standard error is:

$$\text{SE}_{<<F_H>>}(y_c) = \sigma_{F_H}(y_c)/\sqrt{m} \tag{A9}$$

where $F$ is any considered time-dependent 2D field; subscript $H$ refers to the mesoscale component; <<.>> is for averaging in time $t$ (over the study period) and in the along-striation direction $x_a$ (within the tilted boxes on Figures 2 and 3); $y_c$ is the cross-striation coordinate; $\sigma_{F_H}$ is the cross-striation profile of the standard deviation of the mesoscale $F(x_a,t)$ surfaces; and $m$ is the number of independent observations of mesoscale $F(x_a,t)$.

$m$ is estimated as the product of the number $p$ of independent observations of $F$ along the striation axis by the previously obtained $n$ resulting from time-averaging. Based again on the perspective of westward mesoscale eddy propagation, a typical length scale separating two consecutive independent observations is the eddy diameter $D$ = 150 km in the ENP and ESP. Approximate values for $p$ are then obtained by dividing the near-zonal width $L$ of the tilted black boxes on Figures 2 and 3 by $D$ and rounding the result (Table A1). $m$ is in the 100–400 range depending on the tracer field and region (Table A2).

**Table A1.** Computed range for the number $p$ of independent along-striation observations in the ENP (CTZ stands for coastal transition zone) and ESP using the minimum–maximum latitudes and estimated width $L$ of the tilted boxes on Figures 2 and 3, and considering $D$ = 150 km. The values finally retained for $p$ are in brackets.

| Region | ENP | ENP CTZ | ESP |
|---|---|---|---|
| Latitude | 25° N–39° N | 26–42° N | 30° S–44° S |
| $L$ (°) | 20 | 10 | 20 |
| $L$ (km) | 1700–2000 | 800–1000 | 1600–1900 |
| $p$ | 11–13 (12) | 5–7 (6) | 11–13 (12) |

**Table A2.** Computed values for the number *m* of independent observations associated with the cross-striation profiles on Figures 6 and 11. The values for the number *n* of independent observations over the SST, Chl-a (2012–2018), and SSS (2015–2018) records are also indicated.

| Region | ENP | ENP CTZ | ESP |
|---|---|---|---|
| SST | $n = 33, m = 396$ | $n = 33, m = 198$ | $n = 26, m = 312$ |
| SSS | $n = 19, m = 228$ | $n = 19, m = 114$ | $n = 15, m = 180$ |
| Chl-a | | $n = 33, m = 198$ | $n = 26, m = 312$ |

Figures S3–S8 (respectively Figures S9 and S10) are obtained after the application of Equation (A9) and multiplication factor 1.64 to derive 90% confidence intervals for the various mean profiles shown on Figure 6 (respectively Figure 11). The profiles are then considered statistically significant at the 5% level wherever these intervals do not cross the zero line.

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
