# Peer review of "Similarities and Contrasts in Time-Mean Striated Surface Tracers in Pacific Eastern Boundary Upwelling Systems: The Role of Ocean Currents in Their Generation"

_fluids, doi:10.3390/fluids6120455_

Round 1

Reviewer 1 Report

The review is in the attached pdf.

Reviewer 2 Report

This study takes advantage of satellite data and characterized the striations from SST,SSS and chlorophyll. It also compares the original difference of striations in the ENP and ESP regions. They found that in the ENP system, striations from SST and SSS arise from the advection of quasi-zonal jets and striations from chlorophyll is related to transport by mesoscale eddies. Whereas in the ESP region, the striations in SST is induced by eddy transport. This paper also emphasizes the importance of quasi-zonal jets in the striation structures through flow decomposition and advection diagnosis.

Major comments:

  • Diagnosing Eq. (10) is insightful. However, I think it is also important to formulate a balance equation (including advection term) for the tracer striation itself (e.g. SST striation, SSS striation and chlorophyll) and diagnose the advection decomposition of the tracer striation equation. (e.g. maybe the filtered version of the original tracer equation can be useful).
  • As to the advection diagnosis, seems that only horizontal advection is diagnosed. This is understandable, because horizontal advection is more relevant to striations. Yet, it would be worth pointing out the amplitude and structure of vertical advection as well, in order to put the number/structures of horizontal advection in a broader context.
  • Line 566-589。Section 5 summarizes the main conclusion of the paper. It could be improved if you consider adding discussions about the implications of the work, drawback and future work of this study.

Other comments:   

  • Overall, this is a well written paper, however, it could be further improved if the paper emphasizes more about its novel perspectives (both in abstract and conclusion section).
  • Avoid using the jet colorbar in Figs. 1-5, 7, 11 and B11. Also Figs. 7(b),(d)和11(b)do not have colorbar. Maybe it is ignored by the authors? It is not stated clearly in the caption, either.
  • The following writing sounds like oral English and should be improved:Line 63:“just to name a few”,Line 123:“Whenever possible and unless stated otherwise”,Line 275:“from left to right”。
  • Part of the captions of Figs. 2 and 3 are identical. Could we simplify the caption of Fig. 3?
  • Line 272: V in Eq. (1) should be in bold font.
  • 5: Note that panels (d), (g) and (j) are exactly the same as panels (f), (i) and (l). Could we remove one copy of the three panels. Or did you put in the wrong panels?
  • Line 284。 (2)-(4) and Eqs. (A1), (A3), (A4) and (B1) are not displayed properly.
  • Please define ∇ and ∇H in Eqs. (1) and (2). (
  • Line344。Since there are more than one box in each panel of Fig. 2. In the caption of Table 1, you could point clearly which box you are refering to. Is it the black dashed box?
  • Are Lines 487-490 consistent with Figs. 2(b) and (d)?

Reviewer 3 Report

Please see attached file with all my comments on this manuscript

Round 2

Reviewer 1 Report

I am grateful to the authors for their response to the comments. I read the revised manuscript and the authors' response to my comments. The authors have addressed all my comments in detail and have made appropriate modifications in the revised version. I am satisfied with the revised version. Thus, I recommend the manuscript for publication.

Author Response

Once again, we are grateful to the reviewer for his/her careful reading and appreciation of our manuscript, which has been invaluable for the paper improvement.

Reviewer 3 Report

I want to thank the authors for doing good job and adressing my comments.

I have read the authors revisions and I am satisfied with all responses except for my third comment (the use of Fourier tranfrom). The authors might have misinterpreted my comment. I clearly understand the appliying 2-D Hanning window is necessary for subsequent application of the Fourier tranform (periodicity of the function). I also understand that spatial low-pass filtering is nesseseary in order to remove the signal from more energetic large-scale currents. The Fourier spectra of mesoscale fields (after applying low-pass filtering and 2-D Hanning window) contains well-defined spectral peaks. True. However, how do the authors know that all the power within that peak corresponds to striations? If the authors applied EOF decomposion to mesoscale(!) field and report on how many modes apper to be zonally-elongated, and remove those modes from mesosclae(!) field and recomputed the Fourier spectra of newly filtered mesoscale field would probably benefit this study. When computing EOF of mesoscale (!) field one does not need to remove the time mean component. From my previous experience working on a similar topic, the spectral peak observed in mesoscale(!) Fourier spectrum does not entirely correspond to zonally-elongated flow structures. Part of the power within that peak might be due to different flow structutes. I am not insiting on any additional calculations though if the authors think those would be redundant.  I think this is a great study.

The manuscript can be accepted after the above minor revisions.

Author Response

We thank the reviewer once again for his/her careful review and appreciation of our work. The reviewer's comments have been critical in helping us strengthen the paper.

Following the reviewer's further clarification of their third comment, we confirm that we had misinterpreted the reviewer. The point that the reviewer is raising is meaningful. We have thoroughly discussed it among co-authors. Although we agree that the analysis that is being proposed would further strengthen the robustness of the study, we feel that it is not necessary and have decided that we would not include it, for several reasons.

Indeed, we doubt that the FFT results would change much after applying such changes. Visual inspection of the maps on Figures 2 and 3 clearly reveals the striated features in the different mesoscale fields. They are confirmed objectively with the anisotropic ratio analysis. They appear generally consistent with previous published research, including meridional scales and tilt angles from FFT. To us it is quite obvious that the spectral peaks correspond to striations to a large extent, although we cannot completely discard the possibility that the peak may be slightly contaminated with other mesoscale features. Such residual features (assuming they exist) with small zonal wavenumber and with meridional wavenumber broadly consistent with striations (as inferred from the well-defined spectral peak) are therefore likely to be similar to the striations being evidenced in the paper, possibly much weaker, and may therefore be considered as noise according to the framework of our analysis.

On the other hand, if we wanted to address the comment properly, it would require substantial additional work: EOFs would need to be applied to all the fields shown on Fig. 4 and 5 (SST, SSS, Chl-a, SSH, Ug, U), for both domains, and with additional tests to check the sensitivity of the analysis to the chosen geographical domains as recalled by the reviewer. Then the results would have to be reviewed before running all the Fourier analysis again. Of course the figures, tables, and text would need to be adapted. This represents a lot of extra effort for a presumably small improvement (see previous paragraph), and well beyond the definition of minor revision in our opinion.

In addition, even if the purpose of this analysis would be to improve the definition/robustness of the spectral peaks associated with striations, the EOF analysis would potentially reveal additional information about transient striations and/or low-frequency modulation of quasi-stationary striations. As expressed in our previous response and in the revised manuscript, we do not wish to broaden the scope of this already extensive paper to include such information. Yet we would probably not want to ignore it either, if available. This is another reason for refraining from performing the analysis, which is left for future dedicated work as expressed in the revised paper.

We have however included two sentences in the manuscript (2nd revision) to account for the possibility of spectral aliasing (lines 420-425).

Thank you again for helping us to improve the paper.